# Genome-wide association analysis of left ventricular imaging-derived phenotypes identifies 72 risk loci and yields genetic insights into hypertrophic cardiomyopathy

Caibo Ning[1,2,3,11], Linyun Fan[1,2,3,11], Meng Jin[4,11], Wenji Wang[5,11], Zhiqiang Hu[5,11], Yimin Cai [1,11], Liangkai Chen [6,11], Zequn Lu[1], Ming Zhang[1], Can Chen[1], Yanmin Li[1], Fuwei Zhang[1], Wenzhuo Wang[1], Yizhuo Liu[1], Shuoni Chen[1], Yuan Jiang[1], Chunyi He[1], Zhuo Wang[1], Xu Chen[1], Hanting Li[1], Gaoyuan Li[1], Qianying Ma[1], Hui Geng[1], Wen Tian[1], Heng Zhang[1], Bo Liu [4], Qing Xia[5], Xiaojun Yang[7], Zhongchun Liu [8], Bin Li[1], Ying Zhu[1,2,3,9], Xiangpan Li[2], Shaoting Zhang [5,10] ✉, Jianbo Tian [1,2,3,9] ✉ & Xiaoping Miao [1,2,3,9] ✉

Left ventricular regional wall thickness (LVRWT) is an independent predictor of morbidity and mortality in cardiovascular diseases (CVDs). To identify specific genetic influences on individual LVRWT, we established a novel deep learning algorithm to calculate 12 LVRWTs accurately in 42,194 individuals from the UK Biobank with cardiac magnetic resonance (CMR) imaging. Genome-wide association studies of CMR-derived 12 LVRWTs identified 72 significant genetic loci associated with at least one LVRWT phenotype ($P < 5 \times 10^{-8}$), which were revealed to actively participate in heart development and contraction pathways. Significant causal relationships were observed between the LVRWT traits and hypertrophic cardiomyopathy (HCM) using genetic correlation and Mendelian randomization analyses ($P < 0.01$). The polygenic risk score of inferoseptal LVRWT at end systole exhibited a notable association with incident HCM, facilitating the identification of high-risk individuals. The findings yield insights into the genetic determinants of LVRWT phenotypes and shed light on the biological basis for HCM etiology.

The critical role of left ventricular (LV) structure and function in adversely affecting the prognosis of individuals is increasingly acknowledged. Changes in LV structure and function have been associated with acute myocardial infarction[1], heart failure[2] and mortality[3]. Likewise, the LV wall thickness is an independent predictor of morbidity and mortality of hypertrophic cardiomyopathy (HCM)[4], diastolic dysfunction[5], and ventricular arrhythmias[6]. Various LV regional wall thicknesses (LVRWT) occurred with different clinical implications[7,8], and its quantitative analysis is of great significance for the diagnosis of cardiovascular disease (CVD)[9]. Hence, LVRWT

serving as intermediate phenotypes for CVD is needed for timely detection.

Cardiovascular magnetic resonance (CMR) is one of the most popular medical imaging modalities to quantify the LV phenotypes due to its noninvasive and versatile nature[10]. However, the highly variable cardiac structures across different subjects and quantification tasks of small targets pose the biggest challenge to accurate estimations of cardiac wall thicknesses in large investigations[11]. As a result, such a large-scale genome-wide association (GWAS) study of the LVRWT imaging phenotype has not been performed to date. Previous

research on the genetic architecture of CMR-derived traits concentrated primarily on LV volumetric and functional metrics such as stroke volume and ejection fraction[2,12]. To solve this issue, we established a novel deep-learning framework named Myocardial Segmentation and Measurement Method (MSMM) to precisely estimate LVRWT. The UK Biobank (UKB), one of the largest population imaging studies, has acquired both high-quality standardized CMR examinations and genotype data[13], offering a tremendous opportunity to investigate the unknown genetic determinants of LVRWT traits.

In this study, we aimed to investigate the genetic basis of 12 LVRWTs: anterior, anteroseptal, inferoseptal, inferior, inferolateral, and anterolateral LVRWT at end diastole and end systole. Using a deep learning-based method to segment LV structure and design an automatic measurement algorithm to quantify myocardial wall thickness, we performed GWASs on 12 LVRWT traits (all derived from the mid-cavity slice with half obtained during end-systole (ES) phase and half during end-diastole (ED) phase) in 42,194 UKB participants. We then evaluated the putative causal relationships between the LVRWT traits and CVDs using genetic correlation and Mendelian randomization (MR) analyses. Finally, we established the polygenic risk score (PRS) of each LVRWT trait, and validated whether the LVRWT-PRSs have the ability to identify HCM in 439,981 individuals without CMR imaging. Overall, this study substantially enhances our knowledge of the genetic basis of LVRWT and may provide valuable risk stratification guidance to identify high-risk sub-populations in individualized HCM prevention.

## Results

### Segmentation of left ventricle with deep learning

The MSMM procedure we established to measure the thicknesses were presented in Fig. 1. To achieve state-of-the-art performance in segmentation tasks, we trained the deep learning model using the Automated Cardiac Diagnosis Challenge (ACDC) dataset[14,15] annotated by one clinical expert, which has a total of 1902 annotated images. We randomly selected 1420 short axis images for training, 100 short axis images for validation and 382 short axis images for testing. Within the

382 testing images, 80 images belong to the two mid-cavity slices. Since only the mid-cavity slice quantification results were used for the subsequent analysis, we evaluated the methods on two mid-cavity slices of each case to ensure a consistent comparison. Moreover, we included 500 mid-cavity images from the UKB dataset for independent evaluation because that GWASs are performed on the UKB dataset. The Deep Layer Aggregation based deep learning model[16,17] as the network architecture (DLANet) was trained from 1420 ACDC images. The Deep Layer Aggregation (DLA)-based deep learning model as the network architecture was trained from these data. The trained deep learning model was then used to output LV segmentation results in the UKB imaging substudy of over 45,000 people[18]. Detailed information about model training, as well as quantitative and qualitative results, can be found in the "Methods" section and Supplementary Methods.

### Measurement of high-quality left ventricular regional wall thicknesses

We applied a measurement-based quantification method to post-process the deep learning output to measure the LVRWT traits. A total of 45,353 participants with CMR imaging were identified for analyses in the UKB. For these individuals, 6 LVRWT phenotypes at end systole [end-systolic anterior (ES-A), end-systolic anteroseptal (ES-AS), end-systolic inferoseptal (ES-IS), end-systolic inferior (ES-I), end-systolic inferolateral (ES-IL), end-systolic anterolateral (ES-AL) LVRWT] and 6 LVRWT phenotypes at end diastole [end-diastolic anterior (ED-A), end-diastolic anteroseptal (ED-AS), end-diastolic inferoseptal (ED-IS), end-diastolic inferior (ED-I), end-diastolic inferolateral (ED-IL), end-diastolic anterolateral (ED-AL) LVRWT] were available. Following the exclusion of poor image quality and sample quality-control procedures, 42,194 individuals free from a diagnosis of myocardial infarction or heart failure remained (Supplementary Fig. 1). The average age of the cohort at the time of imaging visit was 64.1 years, and 47.3% were men at end systole and 47.2% at end diastole (Supplementary Table 1). The mean and standard deviation of 12 LVRWT phenotypes are reported in Supplementary Table 2. We next sought to estimate the

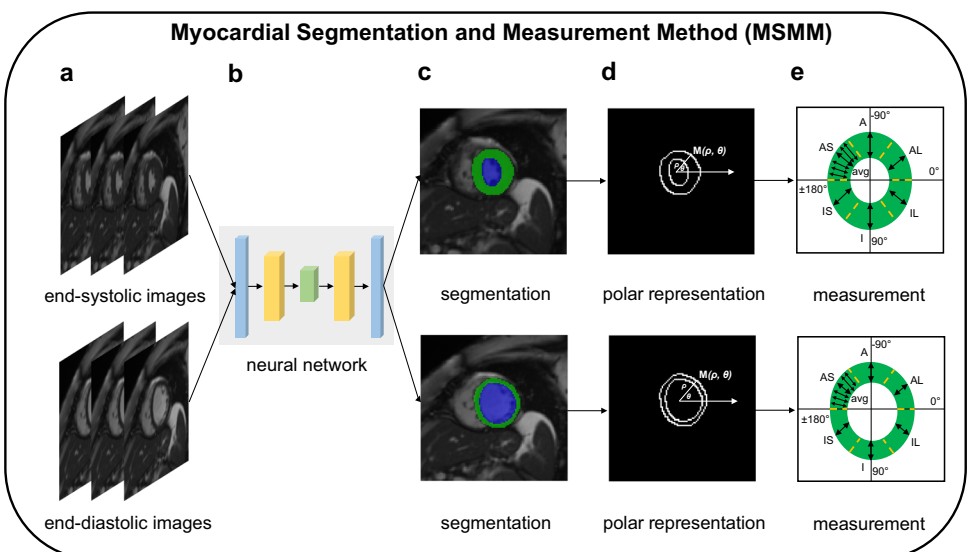

**Fig. 1 | Left ventricular wall thicknesses measurement with deep learning.** A complete and novel framework named Myocardial Segmentation and Measurement Method (MSMM) for the quantification of myocardial wall thicknesses. **a** The original CMR images for end systole and end diastole. **b** The segmentation network architecture is based on the Deep Layer Aggregation (DLA). **c** Deep learning model that has been trained segments the initial CMR images, producing pixel-by-pixel output. Blue: left ventricular cavity, green: left ventricular myocardium. **d** The polar representation of the segmentation contour is used to approximation an underlying function of direction angle mapped to distance with the key points for the left ventricular structure. **e** The function is applied to a set of uniformly spaced direction angles in the neighborhood of the target direction and take the average for a robust quantification. IS inferoseptal, I inferior, IL inferolateral, AL anterolateral, A anternor, AS anteroseptal.

phenotypic correlations of LVRWT traits. The strong positive phenotypic correlations were observed between regions of the same cardiac cycle phases ($r^2 = 0.82$ between ES-IL and ES-AL), and the end-diastolic LVRWT phenotypes and end-systolic LVRWT phenotypes were modestly positively correlated with one another (Supplementary Fig. 2a), which is not unexpected, given their firm physiological associations. Furthermore, to gain insight into the dimensionality of the data, we conducted a principal component analysis (PCA) and identified that utilizing five principal components explained over 90% of the variability in the LVRWT measurements (Supplementary Fig. 2b).

## Genome-wide association studies of left ventricular regional wall thicknesses

To understand the common genetic basis for variation in myocardial wall thickness, we performed a series of GWAS on 12 LVRWT traits within each cohort. Firstly, due to the nonnormal distribution, we performed the rank-based inverse normal transformation of the residuals of LVRWT phenotypes (Supplementary Fig. 3). There was no evidence of confounding from population stratification or cryptic relatedness in our GWAS analyses, as demonstrated by low genomic inflation factor ($\lambda = 1.050$–$1.099$, Supplementary Fig. 4). Furthermore, we identified a total of 72 genome-wide significant variants associated with the 12 LVRWT traits ($P < 5 \times 10^{-8}$), including 16 variants for IS, 2 variants for I, 7 variants for IL, 10 variants for AL, 3 variants for A and 7 variants for AS at end systole stage; 5 variants for IS, 3 variants for I, 3 variants for IL, 4 variants for AL, 3 variants for A and 9 variants for AS at end diastole stage, respectively (Fig. 2 and Supplementary Data 1). Interestingly, there have 10 genetic variants associated with multiple LVRWT traits, resulting in 62 unique variants associated with the 12 LVRWT traits; for example, rs7928419 in SPI1 at 11p11.2 reached genome-wide significance in ES-IS, ED-IS and ED-AS ($P_{ES\text{-}IS} = 1.0 \times 10^{-12}$, $P_{ED\text{-}IS} = 1.3 \times 10^{-9}$, $P_{ED\text{-}AS} = 1.2 \times 10^{-8}$, Supplementary Fig. 5). Furthermore, following the five PCs correction for multiple tests, we observed that 47 variants remained significant ($P < 1.0 \times 10^{-8}$, Supplementary Data 1).

## Functional characterization for risk variants of 12 LVRWT traits

The LVRWT GWAS loci harbored a total of 6345 candidate variants in linkage disequilibrium (LD, $r^2 > 0.6$) with 62 lead variants. To characterize the features for risk variants of 12 LVRWT traits, we first generated the control variants set and functionally defined the genomic position distribution of variants via SnpEff[19]. Compared with control variants, we found risk variants were significant enriched within intron region, gene upstream and downstream regions, and intergenic region (Fig. 3a, b). We next investigated whether risk variants are enriched among genetic regulatory elements. We observed significant enrichments for risk variants of 12 LVRWT traits within the histone marks of promoters [H3K4 trimethylation markers (H3K4me3)], enhancers [H3K4 monomethylation marks (H3K4me1), H3K27 acetylation marks (H3K27ac) and H3K36 trimethylation markers (H3K36me3)], and transcriptional factor binding sites (TFBSs), while significant depletion among repressive transcription [H3K9 trimethylation markers (H3K9me3)] (Fig. 3c). Furthermore, to examine the characteristics for risk variants of 12 LVRWT traits in the context of their potential contribution to CVDs susceptibility, we conducted enrichment analyses using the data of GWAS summary statistics from 11 selected CVDs traits (Supplementary Data 2). As a result, risk variants of 12 LVRWTs were observed significantly enriched among CVDs GWASs loci (Fig. 3d). To sum up, these results indicated that risk variants may regulate the expression of genes by activating chromatin state, and might result in an increased CVDs incidence.

## Identification and functional annotation of susceptible genes associated with left ventricular regional wall thicknesses

We further sought to identify candidate genes influencing LVRWT phenotypic variation using an integrative approach supported by multiple lines of evidence (Supplementary Fig. 6). Based on the downstream analysis of the discovery GWAS summary statistics, for end diastole and end systole, respectively, 31 genes and 21 genes were identified by position (±1 Mb of the lead variant) (Supplementary Data 1); 58 genes and 50 genes were discovered by expression quantitative trait locus (eQTL) mapping (Supplementary Data 3); and 45 genes and 25 genes were identified by transcriptome-wide analysis (TWAS) (Supplementary Data 4–9). Besides, Multi-marker Analysis of GenoMic Annotation (MAGMA) gene-based analyses yielded 55 and 44 significant genes for end diastole and end systole, respectively, (mean $\chi^2$ statistics, $P < 2.64 \times 10^{-6}$) (Supplementary Data 10). We combined all of genes annotated using the four methods. As several loci were shared by multiple traits; counting each locus only once, we totally identified 127 candidate genes at end systole and 95 candidate genes at end diastole, respectively (Supplementary Data 11). Notably, ALPK3 was annotated by the four methods, and rs3803405 in proximity to ALPK3 was the most significant variants in this study. MYBPC3, which was identified by the MAGMA analysis for inferoseptal LVRWT at end systole, is one of the two most well-known HCM causal genes[20]. ALPK3[21], NMB[22], and WNT3[23] loci, which are significant candidate genes that have been linked to inherited CVDs, were shared across most LVRWT traits (Supplementary Fig. 7).

To further dissert the potential function of the candidate genes, we characterized the biological pathways associated with the 127 genes at end systole and 95 genes at end diastole. The GO analyses demonstrated that the candidate genes are remarkably enriched in the biological pathways correlated with the heart development, heart contraction, and cardiac muscle cell development, which are essential for cardiac remodeling (Fig. 4a, b and Supplementary Data 12). The tissue expression analyses were performed separately for 54 specific tissue types, and tissues from the heart atrial appendage and heart left ventricle showed the most significant $P$ value ($P = 2.64 \times 10^{-6}$) in most LVRWT traits (Fig. 4c, d and Supplementary Data 13). In summary, these findings significantly advance our understanding of the genetic basis of LVRWT phenotypes and suggest that candidate genes encode essential proteins involved in the construction and maintenance of sarcomeric infrastructure.

## Heritability and genetic correlation of left ventricular regional wall thicknesses

We next estimated the proportion of LVRWT phenotypic variation attributable to the genotypes by the variance component analyses. The highest genome-wide variant heritability estimates were observed for inferoseptal LVRWT at end systole (ED-IS, at 28%), followed by inferoseptal LVRWT at end diastole (ED-IS, at 25%), while heritabilities were lower for anterior and anterolateral LVRWT at end diastole, which had a heritability of 17% (Fig. 5a). We observed significantly genetic correlations between LVRWT traits ranging from high ($r_g = 0.97$ between ES-IL and ES-A) to low ($r_g = 0.42$ between ES-A and ED-AS; Fig. 5a). Strong positive genetic correlations were generally found between regions of the same cardiac cycle phases and moderate correlations between regions of the different cardiac cycle phases, which may reflect genetic effects acting on the development of the cardiac wall.

Due to the interdependent nature of LVRWT and LV chambers and the importance of wall thickness remodeling, we first sought to quantify the strength of their genetic correlations by linkage disequilibrium score regression (LDSC) analyses using our LVRWT summary statistics and the summary data from a recently published LV GWAS[12]. The LVRWT traits had highly positive genetic correlations with LV volumetric and functional traits such as LV ejection fraction (LVEF) and stroke volume (SV) ($r_g$ range: 0.16–0.60), and negative genetic correlations with LVRWT traits were observed between LV end-diastolic volume (LVEDV), LV end-systolic volume (LVESV) and the body-surface-area indexed versions of these traits, including LVEDVi

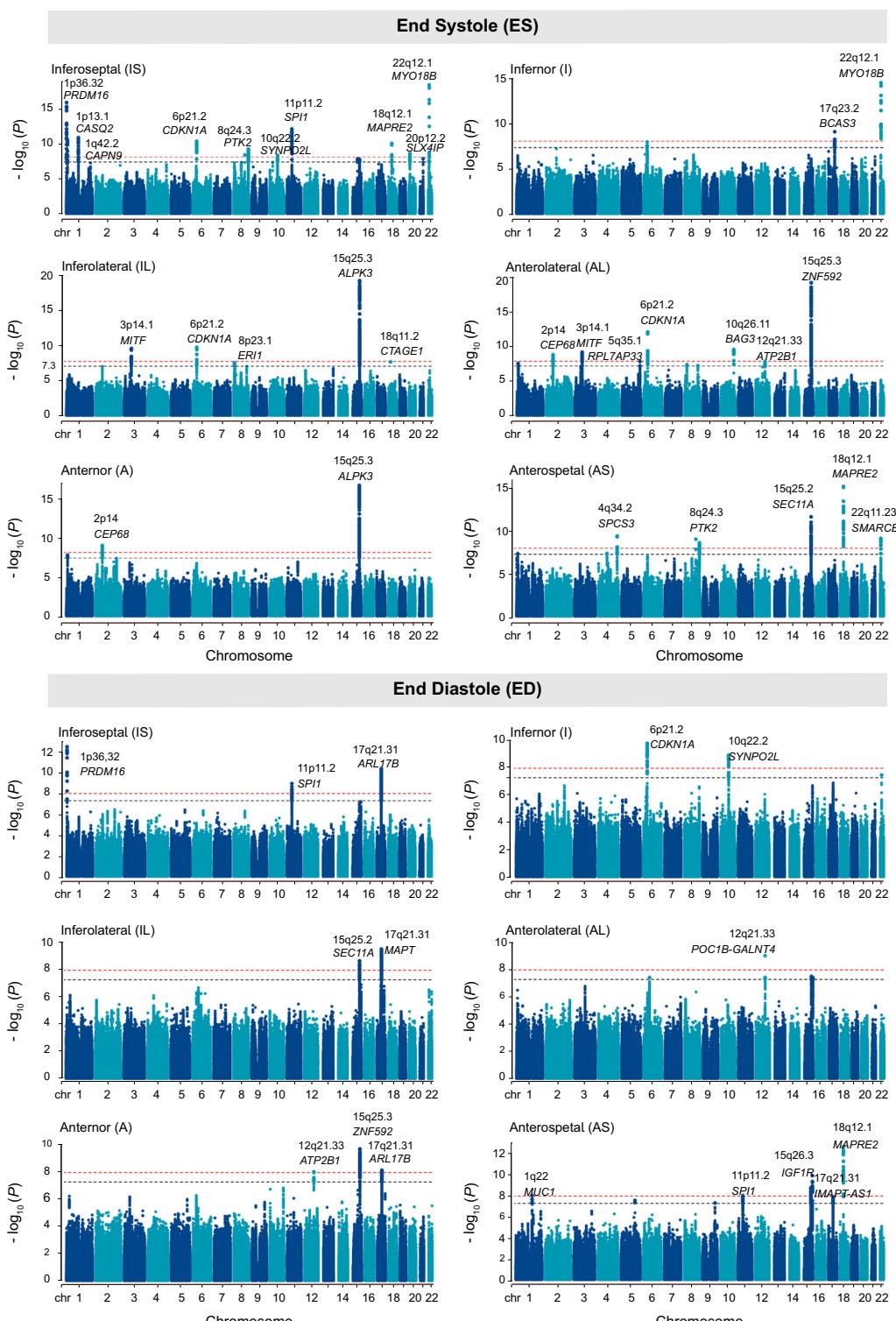

**Fig. 2 | Manhattan plots of genome-wide association studies results for 12 LVRWT phenotypes.** Manhattan plots show the chromosomal position on the *x*-axis and the −log10(*P*) on the *y*-axis for each LVRWT phenotype. The black dashed line indicates the genome-wide significance threshold at $P < 5 \times 10^{-8}$, while the red dashed line represents the significance level after multiple corrections ($P < 1.0 \times 10^{-8}$). Loci that contain variants with $P < 1 \times 10^{-8}$ were labeled with the name of the nearest gene. *P* values are two sided based on the chi-squared test statistics in BOLT-LMM software. LVRWT LV regional wall thickness, IS inferoseptal, I inferior, IL inferolateral, AL anterolateral, A anternor, AS anteroseptal.

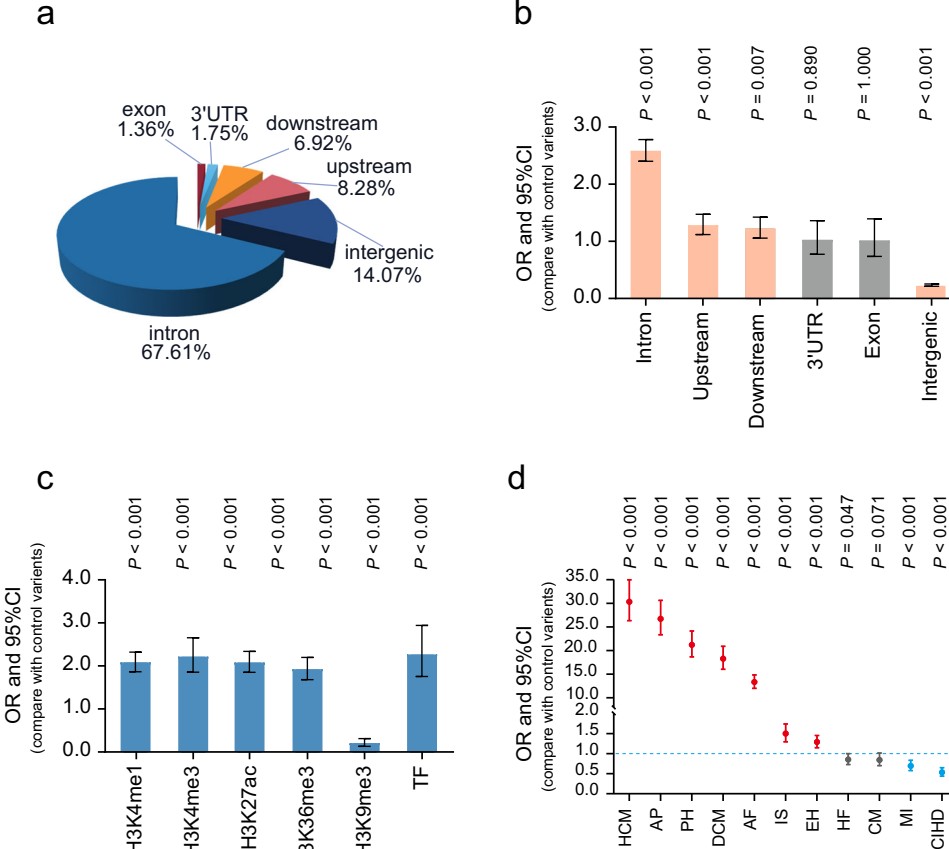

**Fig. 3 | Functional characterization for risk variants of 12 LVRWT traits. a** Pie chart represents the proportions for risk variants of 12 LVRWT traits annotated with each functional category (intron region, gene upstream and downstream regions, intergenic region, 3'-untranslated region (UTR), 5'-UTR and exon region).
**b** Enrichment analyses for risk variants of 12 LVRWT traits in each different functional type of variants. *P* values were calculated by two-tailed Fisher's exact test and bars indicate 95% confidence intervals (CIs). **c** Enrichment analyses for risk variants of 12 LVRWT traits among epigenomic marks, such as H3K4 monomethylation

marks (H3K4me1), H3K4 trimethylation marks (H3K4me3), H3K27 acetylation marks (H3K27ac), H3K36 trimethylation marks (H3K36 me3), H3K9 trimethylation markers (H3K9me3) and transcriptional factor binding sites. *P* values were calculated by two-tailed Fisher's exact test and bars indicate 95% CIs. **d** Enrichment analyses for risk variants of 12 LVRWT traits among 11 CVDs risk loci. *P* value was calculated by two-tailed fisher's exact test. OR is depicted on the *y* axis and bars indicate 95% CIs.

and LVESVi ($r_g$ range: −0.58 to −0.13), which emphasized their strong physiological connections (Fig. 5b and Supplementary Data 14). Additionally, we also investigated the genetic correlations between the 12 LVRWT traits and cardiac measurements that could mediate the CVDs progression. The LVRWT traits were highly positively connected with traits such as heart rate, diastolic and systolic blood pressure ($r_g$ range: 0.11–0.45, Fig. 5b and Supplementary Data 14). Collectively, these findings support the justification for investigating the genetics of LVRWT traits as a complementary gateway to understanding the drivers of cardiac remodeling.

### Genetic correlation analyses between left ventricular regional wall thicknesses with cardiovascular diseases

To examine shared genetic effects between LVRWT traits and CVDs, we performed genetic correlation analyses with GWAS summary statistics from 11 selected CVDs traits. As expected, we observed statistically positive genetic correlations of 12 LVRWT traits with CVDs such as HCM, hypertension, pulmonary hypertension, chronic ischemic heart disease, and ischemic stroke. Anterior LVRWT at end diastole had the highest genetic correlation estimate with HCM than other LVRWT traits (ED-A: $r_g = 0.65$, $P = 7.0 \times 10^{-4}$). For inferoseptal LVRWT, we observed positive genetic correlations with HCM at both end systole and end diastole (ES-IS: $r_g = 0.52$, $P = 1.4 \times 10^{-3}$; ED-IS: $r_g = 0.42$, $P = 6.8 \times 10^{-3}$). Furthermore, HCM as the disease had the highest

genetic correlation estimates with LVRWT than other CVDs ($r_g$ range: 0.37–0.65) (Fig. 5c and Supplementary Data 15). In full, these findings of the genetic correlations between LVRWT traits with CVDs provide quantitative support for genetic overlap.

### Mendelian randomization of left ventricular regional wall thicknesses exposures and cardiovascular diseases

The genetic correlations between 12 LVRWT traits and CVDs risk led us to the hypothesis that the increased LVRWT traits are causally associated with CVDs. We then tested such potential causality between 12 LVRWT exposures and CVDs outcomes using two-sample Mendelian randomization (MR). Although heterogeneity and the number of variants in the exposure-outcome effects are limitations, the findings of the inverse-variance-weighted method and sensitivity analyses support causal relations between 10 of 12 LVRWT traits and increased HCM risk (beta range: 0.45 to 3.10, $P < 0.01$) (Fig. 6 and Supplementary Data 16). We also found robust support for the causal effects of inferolateral, anteroseptal, and anterior LVRWT on cardiomyopathy at end systole and end diastole ($P < 0.01$). However, there is no statistically significant effect on hypertension, angina pectoris, myocardial infarction, chronic ischemic heart disease, and ischemic stroke ($P > 0.05$). Taken together, these findings suggest that the genetic relationships between LVRWT traits and HCM may partly reflect underlying causal processes.

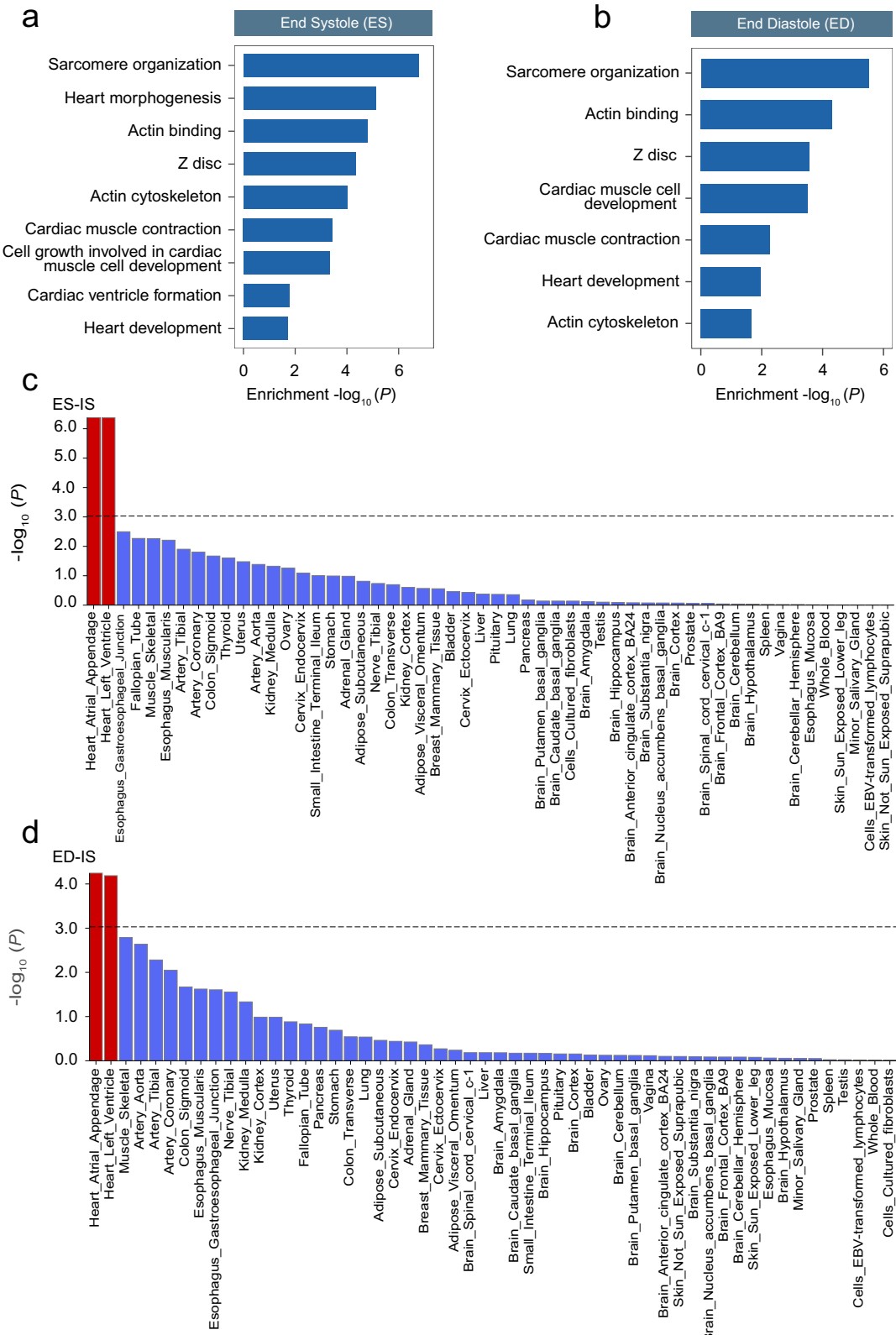

**Fig. 4 | Pathway enrichment and tissue enrichment for susceptible genes associated with LVRWT. a**, **b** Pathway enrichment analyses of Gene Ontology terms were performed at http://kobas.cbi.pku.edu.cn. Significantly enriched GO terms (*P* < 0.05, two-sided) were identified from the analyses of significant genes for end systole (**a**) and end diastole (**b**). **c**, **d** Tissue expression results for 54 specific tissue types were obtained from GTEx v8 using FUMA. The *P* values were calculated as two-sided. those bar that survived multiple correction (*P* < 0.05/54) are denoted with red. Tissue enrichment analysis was performed for inferoseptal (IS) at end systole (ES) in (**c**) and end diastole (ED) in (**d**). IS inferoseptal, ES end systole, ED end diastole.

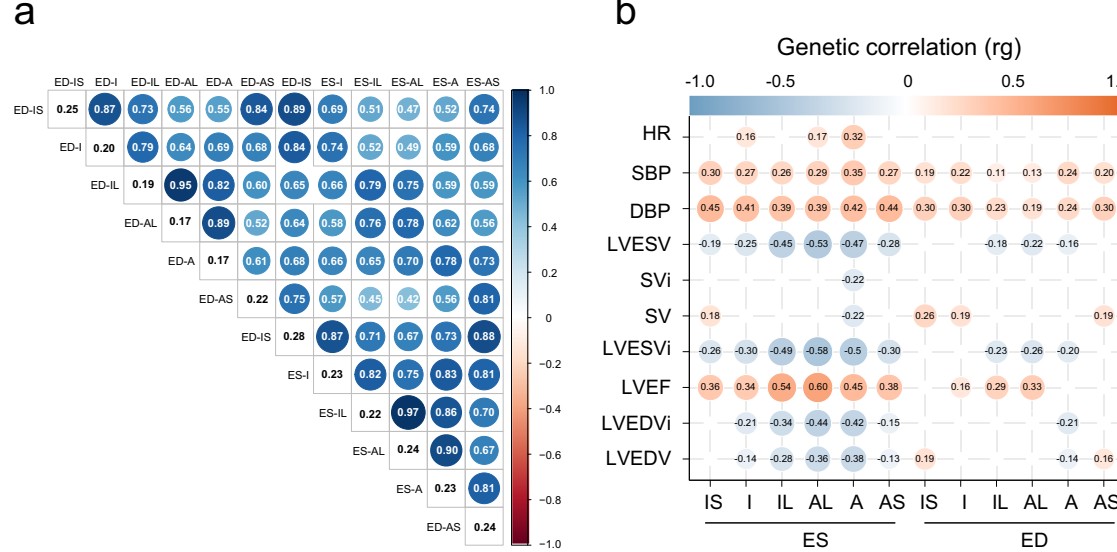

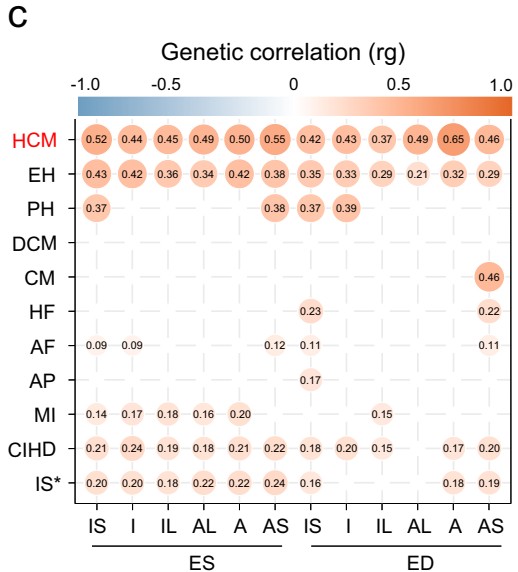

**Fig. 5 | SNP heritability and genetic correlations. a** Genetic correlations between 12 LVRWT traits. Degree of correlation is indicated by color legend and circle area, ranging from −1 to +1. SNP heritability is shown in the diagonal. We estimated the single nucleotide polymorphism heritability ($h^2_g$) and the genetic correlations using BOLT-RELM and LDSC, respectively. Two-sided *P* values shown unadjusted are estimated using LDSC for genetic correlation. Only significant correlations (*P* < 0.05) are shown. **b** Genetic correlations between 12 LVRWT traits and 10 cardiac structure and function traits. Degree of correlation is indicated by color legend and circle area, ranging from −1 to +1. Two-sided *P* values shown unadjusted are estimated using LDSC for genetic correlation. Only significant correlations (*P* < 0.05) are shown. **c** Genetic correlation between the 12 LVRWT traits and 11 CVDs. Degree of correlation is indicated by color legend and circle area, ranging from −1 to +1. Two-sided *P* values shown unadjusted are estimated using LDSC for

genetic correlation. Only significant correlations (*P* < 0.05) are shown. LVRWT LV regional wall thickness, CVDs cardiovascular diseases, IS inferoseptal, I inferior, IL inferolateral, AL anterolateral, A anterior, AS anteroseptal, ES end systole, ED end diastole, EH essential hypertension, PAH pulmonary arterial hypertension, DCM dilated cardiomyopathies, HCM hypertrophic cardiomyopathy, CM cardiomyopathy, HF heart failure, AF atrial fibrillation, AP angina pectoris, MI myocardial infarction, CHID chronic ischemic heart disease, IS* ischemic stroke, HR heart rate, SBP systolic blood pressure, DBP diastolic blood pressure, Lvesv left ventricular end-systolic volume, SVi stroke volume (BSA-indexed), SV stroke volume, LVESVi left ventricular end-systolic volume (BSA-indexed), LVEF left ventricular ejection fraction, LVEDVi left ventricular end-diastolic volume (BSA-indexed), LVEDV left ventricular end-diastolic volume.

## Polygenetic risk scores influence the risk for incident hypertrophic cardiomyopathy

For each of the 12 LVRWT traits, we derived PRSs weighting the genetic dosage by the effect size of independent genetic variants located at autosome ($P < 1 \times 10^{-5}$) from each LVRWT traits GWAS (Supplementary Data 17). We investigated whether PRSs could discriminate the 12 LVRWT traits. As expected, participants with a higher PRS tended to have thicker LVRWT for all 12 LVRWT traits, especially for inferoseptal

LVRWT at end systole and end diastole (Fig. 7a, b and Supplementary Figs. 8a, b–12a, b), suggesting that the PRSs yielded favorable discrimination for the LVRWTs.

Having established the causal effects of LVRWT traits on HCM, we then sought to evaluate whether a genetic predisposition to LVRWT traits is associated with incident HCM in the remaining 439,981 individuals without CMR imaging data. We found that the PRS tertiles of inferoseptal LVRWT at end systole were associated with higher risk of

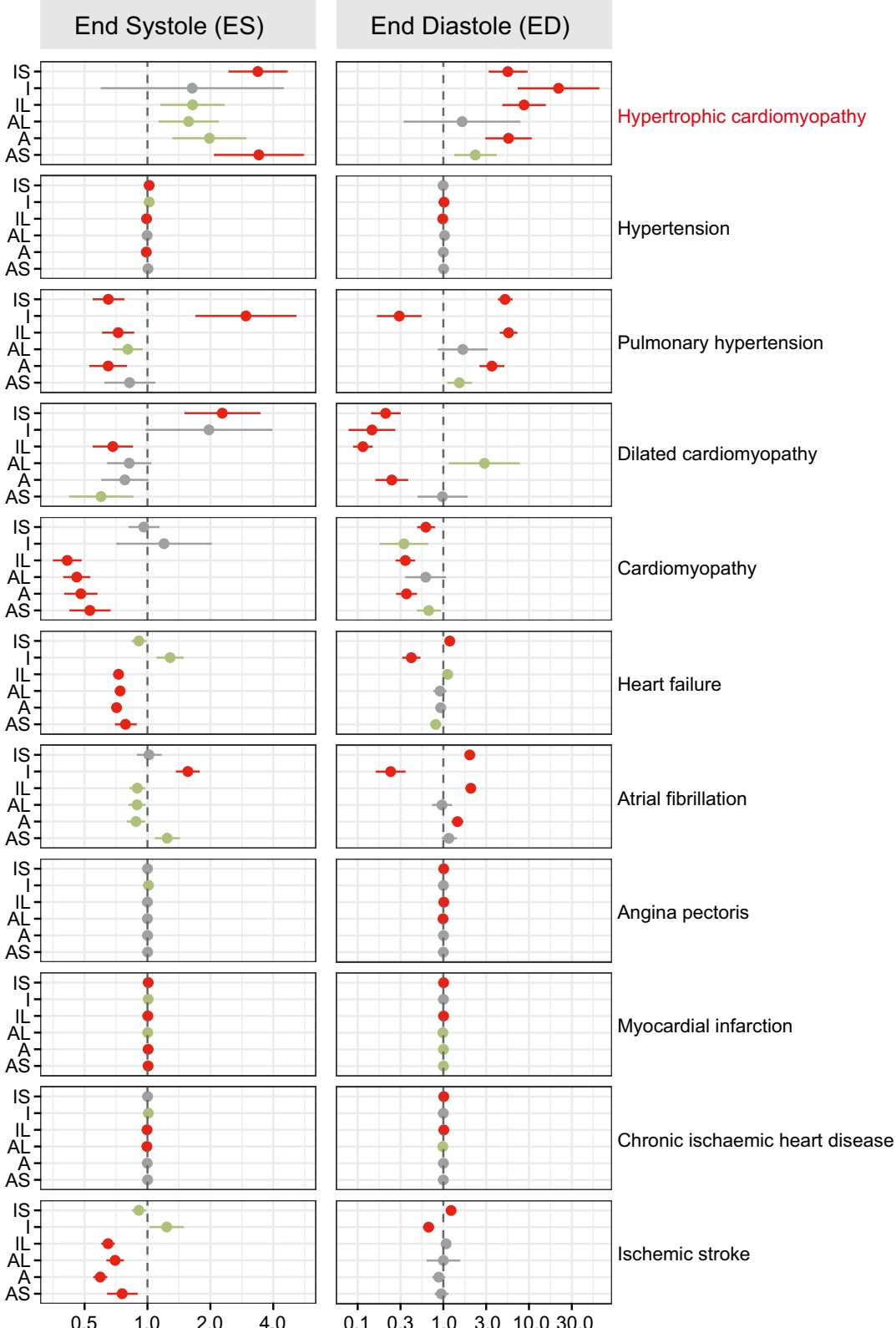

**Fig. 6 | Causal effects between LVRWT phenotypes and CVDs using Mendelian randomization.** The GWAS data sources used for Mendelian randomization are detailed in Supplementary Data 2. Effect sizes are presented as ORs per standard deviation increment in LVRWT in relation to the risk of CVDs. The horizontal bars represent 95% confidence intervals (95% CIs), with color-coding indicating significance levels: red denotes multiple correction for significance (*P* < 0.05/55), green signifies nominal significance (*P* < 0.05), and faded gray indicates non-

significant correlations. For two sample MR analyses, we used MR-Egger regression, weighted median and mode-based estimations as sensitivity analyses, along with the inverse-variance weighted technique as our major model. LVRWT LV regional wall thickness, CVDs cardiovascular diseases, IS inferoseptal, I inferior, IL infer-olateral, AL anterolateral, A anternor, AS anteroseptal, OR odds ratio, SD standard deviation. The dot refers to each variant's mean value of association estimate with exposure and outcome, while the cross symbol represents SE.

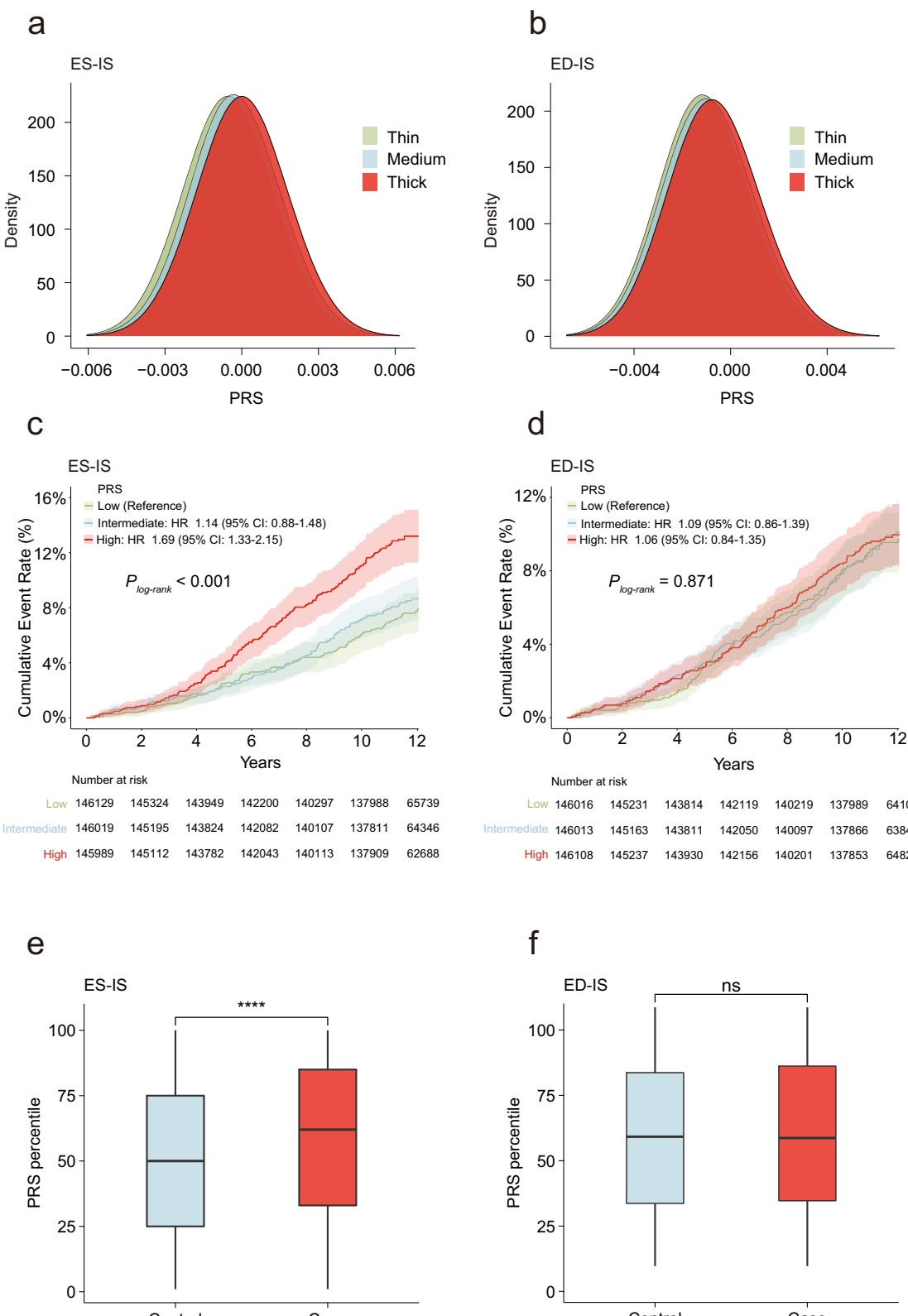

HCM (hazard ratio (HR)$_{ES-IS}$ = 1.69, 95% CI = 1.33–2.15, $P = 2.31 \times 10^{-5}$ for high tertile vs. low tertile, Supplementary Data 18). As expected, the PRS of inferoseptal LVRWT at end systole exhibited effective risk stratification within the HCM population (Fig. 7c). Moreover, we found that the PRS of inferoseptal LVRWT at end systole enabled a clear distinction between incident HCM cases and healthy controls (Fig. 7e). In addition, we found the similar results in inferolateral, anterolateral,

anterior and anteroseptal LVRWT at end systole, and inferior, inferolateral and anterior LVRWT at end diastole (Supplementary Figs. 8d, f, 9c–f, 10c, e, 11c–f and 12c, e). Regrettably, the other 4 LVRWT traits (inferior LVRWT at end systole, and inferoseptal, anterolateral, and anteroseptal LVRWT at end diastole) did not exhibit strong performance in identifying individuals at elevated risk (Fig. 7d, f and Supplementary Figs. 8c, e, 10d, f and 12d, f). We also assessed the

**Fig. 7 | Distribution of PRS and cumulative incidence of HCM stratified by PRS. a, b** Distribution of PRS for participants with thin, medium and thick LVRWT. PRS of inferoseptal LVRWT end systole (**a**) and end diastole (**b**) yielded discrimination for the LVRWTs. **c, d** A total of 439,981 individuals unrelated to the CMR cohort. Those in the first tertiles of genetically predicted inferoseptal LVRWT are depicted in green, the second tertiles are depicted in blue and the last tertiles are depicted in red. The darker shades represent the central estimate of the cumulative incidence (defined as 1-the Kaplan–Meier survival estimate). The lighter shades represent the respective 95% CIs. The x axis depicts years since enrollment in the UKB; the y axis depicts cumulative incidence of HCM. Strata based on genetic prediction of inferoseptal LVRWT at end systole (**c**) and end diastole (**d**). Distribution of ES-IS (**e**) and ED-IS (**f**) PRS percentiles for HCM cases (n = 420) and controls (n = 439,561). For all box plots: central line of each box, median; top and bottom edges of each box, first and third quartiles; whiskers extend 1.5× the interquartile range beyond box edges. P values were calculated by a two-sided Student's t test. Asterisk denotes statistically significant differences *p < 0.05; **p < 0.01; ***p < 0.001. PRS polygenic risk score, HCM hypertrophic cardiomyopathy, LVRWT LV regional wall thickness, ES end systole, ED end diastole, IS inferoseptal.

correlation between the constructed PRS for LVRWT traits and 10 other CVDs (Supplementary Data 18). We calculated the C-statistic for our HCM PRSs and noticed that it is not sufficiently satisfactory. However, a slight improvement is noted after further integrating the relevant PRS with clinical risk factors, as depicted in Supplementary Fig. 13. Despite the modest C-statistic values, this enhancement underscores the efficacy of our PRS in identifying individuals at an elevated risk for HCM. In total, higher genetically determined LVRWT was associated with higher HCM risk, which may provide valuable risk stratification guidance to identify high-risk individuals.

## Discussion

To our knowledge, this study is the first largest individual-level GWAS to investigate the genetic architecture of LVRWT traits. We established a novel deep learning algorithm to assess 12 LVRWTs accurately using CMR imaging in a large population-based biobank. We identified 72 significant genetic loci associated with at least one LVRWT phenotype, and candidate genes were actively participating in heart development and heart contraction pathways. Furthermore, we evaluated the causal relationships between the LVRWT traits and CVDs with MR analyses and found supporting evidence of LVRWT being causal for increased HCM. We demonstrated that PRSs derived from LVRWT traits are associated with incident HCM. This suggests that the genetic basis of LVRWT traits may offer valuable information for the risk stratification of individuals, aiding in the early screening of HCM (Supplementary Fig. 14).

Due to various subjects and cardiac disorders having varied forms and structures, and the LV myocardium experiencing complicated regional deformation during the systole and diastole phases of the cardiac cycle[11], it is challenging to obtain an accurate estimation of LVRWT. Therefore, few studies were carried out on the quantification of LVRWT, and little was known about the heritability and genetic basis of LVRWT. Deep learning is a powerful tool for deriving quantitative phenotypes from raw signal data at a population level. In this paper, by using a deep learning-based method to segment LV structures, we were able to design an automatic measurement algorithm to quantify myocardial wall thicknesses accurately. Reassuringly, the mean absolute errors of LVRWT of our solutions are at most 1.04 mm, suggesting that the MSMM has the highest accuracy of LVRWT. Compared to the maximum wall thickness or mean wall thickness[24,25], we finely divided the wall thickness into 12 LVRWTs, permitted characterization of the genetic architecture of each region and provided new biological insights. According to heritability estimates ranging from 17% to 28%, our research revealed that a sizable portion of the LVRWT phenotypic variability is explained by the underlying genetics. Furthermore, the modest genetic correlations and limited locus overlap of the 12 LVRWT traits highlighted their distinct biology.

A total of 6345 candidate variants are identified in 62 unique loci. Characterizing the genomic characteristics for these risk variants of 12 LVRWT traits allowed us to identify their enrichment in intron region, gene upstream and downstream regions, and intergenic region, which sheds light on the regulatory functions of risk variants[26]. This notion is also supported by the fact that candidate variants are enriched in TFBSs and histone modification markers like H3K4me3, H3K4me1, H3K27ac, and H3K36me3, which suggests a connection between chromatin status and regulatory properties of risk variants of LVRWT traits. Importantly, we further confirmed a significant enrichment of these risk variants in the CVDs GWAS loci. This finding strengthens the growing evidence that the majority of GWAS risk variants influence disease risk via their regulatory activities.

Extensive multilayered bioinformatic annotations identified candidate genes that have an important role in cardiac muscle cell development and heart contraction. In light of recent Khurshid's study, it conducted a GWAS of CMR-derived left ventricular mass indexed (LVMI) in the UKB. By comparing the loci of the 13 genes from Khurshid's study, we found an overlap with 5 genes also identified in our research[27]. This overlap underscores the consistency of genetic associations related to cardiac traits and cardiomyopathy between our study and Khurshid's study. In addition, many of candidate genes identified in this study have previously been linked to CVDs. Specifically, many of candidate genes overlapped with previously reported loci for the cardiac volumetric and functional phenotypes, such as *ALPK3*[28], *NMB*[29], *CASQ2*[30], *CDKN1A*[31], *ACTN2*[32] and *GATA4*[33]. For example, the alpha kinase 3 (*ALPK3*) encodes a kinase previously implicated as the causal gene in HCM[21]. *ALPK3* regulates the expression of transcription factors (like HEY and HAND proteins) to induce differentiation and maturation of cardiomyocytes at the beginning of the process[34]. Mutations in *ALPK3* cause familial cardiomyopathy and demonstrate loss of function as the underlying genetic mechanism[35]. Additionally, HCM is a disease of sarcomere proteins, which are composed of thick and thin filaments and Z discs[20]. In agreement with this notion, we also discovered loci like *SYNPO2L, GBAP1, MYOZ1, CAPN9, DMTN,* and *MTSS* that had not been identified in earlier research on cardiac volumetric and functional traits. Interestingly, these genes were revealed to actively participate in Z disc, myofilament, sarcomere, and actin pathways. As an illustration, the scaffolding and actin/myosin regulatory protein known as synaptopodin 2 Like (*SYNPO2L*), which is expressed in the cardiac muscle and localizes at the Z-disc and interacts with a number of other actin proteins, has been linked to cardiac arrhythmia[36]. Altogether, several candidate genes are involved in heart development and contraction, and Z disc pathway activation, suggesting that candidate genes may contribute to the development of cardiomyopathies.

As the strong interdependence and genetic correlations between LVRWT and LV volumetric and functional traits, a significant fraction of the LVRWT loci were linked to previously identified loci for LV volumetric and functional imaging traits. This finding supports the justification for investigating the genetics of LVRWT traits as a complementary gateway to understanding the drivers of cardiac remodeling[37]. Previous studies have indicated that LVRWT traits are independent predictors of a number of clinical outcomes such as stroke[38], ventricular arrhythmia[39], and hypertension[40]. Although human measurement of ventricular wall thickness is prone to variability, a multi-center longitudinal cohort study has reported that LVRWT is a key imaging biomarker in HCM, guiding diagnosis, risk stratification, and clinical management[24]. In consistent with these findings, we also systematically examined the genetic relations between LVRWT traits and CVDs obtained from published GWASs. We observed genetic associations between LVRWT traits and CVDs, such as HCM, hypertension, pulmonary hypertension, chronic ischemic

heart disease, and ischemic stroke, which further reinforced the importance of LVRWT in maintaining efficient circulatory physiology. The results of the MR analysis also showed that LVRWT traits are potentially causally linked with CVDs. Notably, LVRWT traits have the strongest causative effects on HCM than other CVDs. Specifically, these findings point to the notion that HCM may represent the extreme of LVRWT phenotypic variations in certain people, which is helpful to demonstrate that LV wall thickening is the most consistent clinical marker of HCM[41].

GWAS has identified genetic variations that contribute to complex traits and diseases, leading to the development of PRS. The effectiveness of the PRS is often evaluated by determining whether it can help screen the high-risk population to guide clinical or personal decision-making[42]. In agreement with this development, our analyses of PRSs shed light on the linkage between genetically determined LVRWT phenotypic variations and the incidence of HCM. Interestingly, the top-performing PRS, which included 115 variants linked to inferoseptal LVRWT at end systole, successfully identifies individuals at high risk of HCM, especially in those without CMR data. In accordance with recommendations, inferoseptal hypertrophy with a thickness greater than 15 mm was a major component of the diagnostic criteria for HCM[43,44]. The PRSs might identify high-risk, asymptomatic individuals who would benefit from CMR to screen for HCM. Likewise, previous studies also have reported that the PRSs of LV structure and function traits predict heart failure events[2] and incident dilated cardiomyopathy[12]. Collectively, PRS may provide complementary information within guideline-supported frameworks to better stratify different trajectories of HCM risk and inform clinical decision-making for primary prevention and early screening.

We acknowledge some limitations in our study. First, our study population largely consisted of European ancestry, limiting generalizability to other populations, and validation of our findings in more diverse cohorts is needed to assess their applicability to non-European populations. Second, further validation of PRS performance is required in independent cohorts. While PRSs derived from LVRWT traits showed associations with an increased risk of HCM, it's important to recognize that the predictive performance of these PRSs were relatively limited. Our future research will focus on identifying additional genetic loci and molecular markers, with the goal of enhancing the predictive capabilities of our models. Furthermore, while our study provides strong statistical support for the loci that are highly specific for LVRWT phenotypic variation, future experimental studies using gene-editing techniques in cellular and animal models are warranted to elucidate the functional roles of the highlighted risk genes and the underlying mechanisms modulating LVRWT. Lastly, although our findings provide valuable insights into the genetic basis of LVRWT, the complexities of genetic regulation and functional mechanisms within specific cardiac cell types might require more advanced methodologies, such as emerging single-cell genomics techniques, to fully unravel the intricate genetic networks underlying LVRWT and related cardiac remodeling.

In conclusion, using a novel deep learning algorithm to quantify LVRWT measured by CMR, we uncover common variants at 72 significant genetic loci associated with LVRWT, and implicate candidate genes linked to HCM. Moreover, we demonstrate that PRSs derived from LVRWT traits are associated with incident HCM, even in individuals without CMR data. Altogether, these findings represent a substantial advance in our understanding of the genetic architecture of LVRWT phenotypes and shed light on the biological basis for HCM etiology, which may lead to potential novel therapeutic targets and personalized risk stratification strategies in the future.

## Methods
### Study population
The UKB is a large prospective cohort study of over 500,000 participants recruited at 22 assessment centers across the UK between 2006 and 2010. It has gathered a wealth of information on participants, including health and lifestyle data, physical measurements, biological samples, imputed genome-wide genotypes and imaging data. All participants provided informed consent. The ethical committees from the North West Multi-Center Research Ethics approved the study.

### Sample selection
Disease information based on the UKB Field ID and date of first inpatient diagnosis is provided in Supplementary Data 19. In total, we analyzed 45,353 participants with CMR who had not withdrawn consent as of December 2020. After excluding participants with poor imaging, missing information on genetic data, previous myocardial infarction, diagnosis of heart failure as well as body mass index (BMI) < 16 or > 40 kg/m², 42,194 individuals were included in the analysis following the quality-control procedures outlined in Supplementary Fig. 1. Among the initial 456,937 individuals without CMR data, we excluded individuals without genotype data, resulting in a final sample size of 442,889 individuals. After accounting for missing information on smoking and alcohol consumption, the sample size was further reduced to 440,085 individuals. Additionally, in subsequent analyses focusing on specific diseases, we further excluded individuals who had the corresponding disease prior to their enrollment in the UKB. For instance, in the case of HCM, we excluded 104 individuals who had HCM before their enrollment, resulting in a final sample size of 439,981 individuals.

### Definitions of covariates and phenotypes
All covariates recorded at the imaging visit were used in the analysis where possible. The UKB Data-Fields of covariates were listed in Supplementary Data 20. BMI was calculated as: weight(kg)/height(m)². The definitions were used for GWAS participant exclusion and PRS assessment.

### Semantic segmentation and deep learning model training
CMR protocol had previously been described in detail[13]. In brief, CMR was performed with 1.5 Tesla scanners (MAGNETOM Aera, Syngo Platform VD13A, Siemens Healthcare, Erlangen, Germany) using electrocardiographic gating for cardiac synchronization. Long-axis cines and a complete short-axis stack of balanced steady-state free precession cines were acquired at one slice per breath hold.

Semantic segmentation aims to assign a label to each pixel in an image. In the proposed MSMM framework, the deep learning model for segmentation learns to extract MR image features and output one of three different classes for each pixel, including background, LV cavity, and LV myocardium. Labels of LV structures in the ACDC (https://www.creatis.insa-lyon.fr/Challenge/acdc/index.html) dataset are annotated by one clinical expert. There are 1420 images for training process, 100 images for validation and 382 images for testing process. Additionally, we resorted to a doctor who has 7 years of clinical experience to label the LV cavity and LV myocardium on the UKB dataset for a comprehensive evaluation. During the testing stage, 80 mid-cavity slices from the ACDC dataset and 500 mid-cavity slices from the UKB dataset were employed for evaluating the performance of the segmentation model.

We utilized the DLANet[16,17] architecture as a segmentation model, which uses two types of structures: the Hierarchical Deep Aggregation (HDA) and the Iterative Deep Aggregation (IDA). With the HDA and IDA, the DLA architecture can better merge information from multiple layers and scales, and achieve state-of-the-art performance in the segmentation task. DLANet with 166 hidden layers and 20,576,932 parameters was implemented in Pytorch. Training and inference processes were performed on NVIDIA GeForce GTX 1080 Ti GPUs. During the training process, the DLANet was trained from scratch for 1000 epochs. We employed the Adam[45] optimizer for minimizing the Dice Loss[46]. The initial learning rate was set to $1 \times 10^{-3}$ and decayed by 0.99

every epoch. The weight decay was $1 \times 10^{-5}$. At the inference stage, one frame of a patient with all slices in the size of $(8-12)*256*256$ will take about 0.5 s on one GPU. Additionally, we compared the performance of the DLANet with state-of-the-art methods by re-implementing them and reporting the results in Supplementary Data 21 and 22. The comparative analysis across different datasets provides valuable insights into the efficacy of our proposed deep learning model.

Before the training process, we first performed data preprocessing operations. The in-plane resolution of the original cardiac MR images was resampled to 1.25 mm × 1.25 mm, and then the resampled images were cropped or padded to a size of 256 × 256. During training, data augmentations were employed including random shifting, rotation, scaling and so on. For images at the base or apex of the LV, we oversampled them more than one times in an epoch to increase their frequency during the training data and improve the performance on these images for their difficulty. We ran validation every 20 epochs of training, and when the training process was complete, we selected the DLANet that performs best on the validation set for testing. Finally, the trained model output LV segmentation results on the UKB dataset and the results were used for subsequent quantification of the myocardium.

For quantification, we designed a measurement-based method to calculate wall thicknesses. Before the measurement, we normalize the MR images[47] by rotation to the fixed arrangement in order to perform subsequent measurements. The thicknesses rely on the calculation of the distance between the center and the contour of the mask, either epicardium or endocardium, for some direction. To this end, we first converted the Cartesian coordinate system into a polar coordinate system and then generated the distance based on the direction angle. Finally, we averaged ten measurements for wall thicknesses. We then used MSMM to calculate the myocardial thicknesses in the UKB. We obtained accurate estimations of 12 LVRWT traits which included 6 LVRWT phenotypes at end systole (ES-IS, ES-I, ES-IL, ES-AL, ES-A, and ES-AS) and 6 LVRWT phenotypes at end diastole (ED-IS, ED-I, ED-IL, ED-AL, ED-A, and ED-AS). Additionally, we assessed the DLANet's performance across various training dataset sizes, as detailed in Supplementary Data 23 and 24, demonstrating the model's robustness even with limited training data. The techniques used to postprocess the deep learning output to measure wall thickness is described in the Supplementary Methods.

### Genotyping and imputation

Detailed information on genotyping and imputation in the UKB has been described previously[48]. Briefly, participants were genotyped based on UK BiLEVE Axiom™ Array by Affymetrix (807,411 markers for 49,950 participants) and UKB Axiom Array by Affymetrix (825,927 markers for 438,427 participants). Genotype imputation was based on merged UK10K sequencing and 1000 Genomes phase3 reference panels with SHAPEIT3 and IMPUTE3[49]. Variant positions were keyed to the GRCh37 *human* genome reference.

### GWAS and heritability

We calculated the residuals by regressing LVRWT phenotypes on the covariates age, sex, BMI, and imaging center. We performed the GWAS of each normalized LVRWT trait using a linear mixed-model method using ~8.5 million well-imputed variants with minor allele frequency (MAF) ≥ 1% and imputation quality (INFO) score >0.3 by BOLT-LMM. We next estimated the heritability explained by the genotyped variants ($h^2_g$ variant) using BOLT-RELM. Both GWAS and heritability analysis models were adjusted for age, sex, BMI, and principal component (PC) 1–10.

### Genetic correlation

Using summary statistics, we applied LDSC software[50] to estimate the genetic correlations (1) between 12 LVRWT traits; (2) between 12 LVRWT traits and 10 cardiac structure and function traits[12]: heart rate, diastolic blood pressure, systolic blood pressure, LV end-diastolic volume (LVEDV), LV end-systolic volume (LVESV), stroke volume (SV), the body-surface-area (BSA) indexed versions of these traits (LVEDVi, LVESVi, and SVi), and LV ejection fraction (LVEF); and (3) between 12 LVRWT traits and 11 CVDs: hypertension, pulmonary hypertension, dilated cardiomyopathy, HCM, cardiomyopathy, HEART FAILURE, atrial fibrillation, myocardial infarction, angina pectoris, chronic ischemic heart disease, ischemic stroke. Data sources of summary statistics for genetic correlation analyses were listed in Supplementary Data 2.

Given the unavailability of GWAS summary data for hypertrophic cardiomyopathy, pulmonary arterial hypertension, and dilated cardiomyopathies, we performed GWAS analyses utilizing ~8.5 million well-imputed variants, each with a MAF ≥ 1%, and an INFO score exceeding 0.3, using data from the UKB. Each GWAS analysis was adjusted for covariates including age, sex, BMI, smoking status, alcohol intake frequency, and the first ten principal components (PC1-10). The sample sizes for each disease were as follows: hypertrophic cardiomyopathy (552 cases and 2208 controls), pulmonary arterial hypertension (2047 cases and 8301 controls), and dilated cardiomyopathies (1303 cases and 5281 controls).

### Functional follow-up with FUMA

The web tool Functional Mapping and Annotation of Genome-Wide Association Studies (FUMA) has previously been described in detail (https://fuma.ctglab.nl/)[51]. Genome-wide significant variants were defined by trait associations $P < 5 \times 10^{-8}$ and $r^2 < 0.1$ with correlated variants in 250 kb region using FUMA.

We utilized two main approaches to map genome-wide significant loci to genes via FUMA default settings and specialized datasets, as described follows: (1) positional mapping of variants, whereby variants within a 10kB window from known protein-coding genes in the *human* reference assembly (GRCh37/hg19) are mapped; (2) eQTL mapping whereby allelic variations at a variant is significantly linked to expression of a gene, where we considered eQTLs within heart atrial appendage and heart left ventricle from GTEx v8.

We also performed a generalized gene-set analysis using MAGMA within FUMA. variants within exonic, intronic, and untranslated regions were chosen for each gene. The 18,888 protein-coding genes were used in MAGMA. The mean of the summary statistic ($\chi^2$) of GWAS for the variants in a gene was used to determine the gene-based $P$ value[52]. The Bonferroni method was used to calculate the $P$ value significance threshold, which is $2.64 \times 10^{-6}$ when 0.05 is divided by the total number of genes (18,888).

### Transcriptome-wide association study

For each of the 12 LVRWT phenotypes, we performed a TWAS to identify the most strongly associated gene at each locus based on imputed cis-regulated gene expression. We used FUSION with eQTL data from GTEx v8. Precomputed transcript expression reference weights for the LV (5886 genes), the atrial appendage (6740 genes), and the artery coronary (3989 genes) were obtained from the FUSION authors' website (http://gusevlab.org/projects/fusion/). A $P < 8.49 \times 10^{-6}$ was considered significant in LV tissue, a $P < 7.42 \times 10^{-6}$ was considered significant in atrial appendage tissue, a $P < 1.28 \times 10^{-5}$ was considered significant in artery coronary tissue. FUSION was then run with its default settings.

### Enrichment and tissue expression analyses

Functional enrichment and pathway characterization of the 127 candidate genes at end systole and 95 candidate genes at end diastole were done in the website (http://kobas.cbi.pku.edu.cn/) to obtain Gene Ontology (GO) terms. Tissue expression analyses were obtained from GTEx which were also integrated in FUMA. Average gene expression per tissue type was utilized as a gene covariate to test for a positive link

between gene expression in a given tissue type and genetic correlations.

## Mendelian randomization

We used MR to estimate causal effects between the 12 LVRWT exposures and 11 CVDs. For two sample MR analyses, we used MR-Egger regression, weighted median and mode-based estimations as sensitivity analyses, along with the inverse-variance weighted (IVW) technique as our major model. To determine whether any discernible influence was mediated by outliers, the MR Pleiotropy RESidual Sum and Outlier (MR-PRESSO)[53] approach was applied. In brief, we selected variants that were genome-wide significant ($P < 5 \times 10^{-8}$) for each LVRWT trait, and remove variants in LD using default settings ($r^2 < 0.001$, kb = 10) to generate independent variations. A $P < 0.05/5$ traits = 0.01 was considered statistically significant. All analyses were performed using the R package TwoSampleMR.

## Polygenic risk score

We used the C + T (clumping + thresholding) method[54] to construct the polygenic risk score (PRS) of each LVRWT trait based on the effect sizes derived from the LVRWT GWASs. The PRS was calculated through a weighted model, as shown below:

$$PRS_j = \sum_{i=1} \beta_i G_{i,j}$$

where $\beta$ values (the log of odds ratio) is the summary statistic for the effective allele and $G$ is the number of the effective allele observed. We used variants with genome-wide significant ($P < 1 \times 10^{-5}$) and clumping window ($r^2 < 0.1$, kb = 250) to derive PRS. We repeated this procedure for each of the 12 traits, producing 12 PRSs. We categorized participants into three genetic risk levels: low (lowest tertile), intermediate (second tertile) and high (highest tertile).

Additional and detailed analyses are available in the Supplementary Methods.

## Reporting summary

Further information on research design is available in the Nature Portfolio Reporting Summary linked to this article.

## Data availability

Data from the UKB (www.ukbiobank.ac.uk/register-apply) are available to all researchers upon making an application. This research has been conducted using the UKB Resource under Application 63454. Data sources of publicly available GWAS results were listed in Supplementary Data 2. The GWAS summary statistics generated in this study have been deposited in the Human Genome Research Institute GWAS Catalog under accession codes: GCST90278508 to GCST90278519 for the 12 LVRWTs, GCST90296096 for HCM, GCST90296097 for dilated cardiomyopathy, and GCST90296098 for pulmonary hypertension. Source data are provided with this paper.

## Code availability

Data processing scripts used to perform the analyses described herein are available at https://github.com/goodlucknb/LVRWT-script.

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

## Acknowledgements

We are grateful to all members who participated in the study, as well as all individuals who helped us complete the research. This work was supported by National Science Fund for Excellent Young Scholars (NSFC-82322058), Program of National Natural Science Foundation of China (NSFC-82103929, NSFC-82273713), Young Elite Scientists Sponsorship Program by CAST (2022QNRC001), National Science Fund for Distinguished Young Scholars of Hubei Province of China (2023AFA046), Fundamental Research Funds for the Central Universities (WHU:2042022kf1205) and Knowledge Innovation Program of Wuhan (whkxjsj011, 2023020201010073) for J.T.; National Science Fund for Distinguished Young Scholars of China (NSFC-81925032), Key Program of National Natural Science Foundation of China (NSFC-82130098), the Leading Talent Program of the Health Commission of Hubei Province, Knowledge Innovation Program of Wuhan (2023020201010060) and Fundamental Research Funds for the Central Universities (2042022rc0026, 2042023kf1005) for X.M.; the Beijing Nova Program (Z201100006820064, Z211100002121165).

## Author contributions

S.Z., J.T. and X.M. were the overall principal investigators in this study who conceived the study and obtained financial support. S.Z., J.T. and X.M. were responsible for the study design and supervised the entire study. C.N., L.F., M.J., W.J.W., Z.H., Y.C. and L.C. organized the data,

carried out the statistical analysis and participated in writing the first draft of the manuscript. C.N. designed and drew the figures. L.F. and M.Z. critically revised the manuscript. Z.Q.L., C.C., Ya.L., F.Z., W.Z.W., Yi.L., S.C., Y.J., C.H., Z.W., X.C., H.L., G.L., Q.M., H.G., W.T., H.Z., B.L., Q.X., X.Y., Z.C.L., B.L., Y.Z. and X.L. organized and analyzed the data. These authors jointly supervised this work.

## Competing interests

The authors declare no competing interests.

## Additional information

[1]Department of Epidemiology and Biostatistics, School of Public Health, Wuhan University, Wuhan 430071, China. [2]Department of Radiation Oncology, Renmin Hospital of Wuhan University, Wuhan 430071, China. [3]Department of Gastrointestinal Oncology, Zhongnan Hospital of Wuhan University, Wuhan 430071, China. [4]Department of Oncology, Tongji Hospital, Tongji Medical College, Huazhong University of Science and Technology, Wuhan 430030, China. [5]SenseTime Research, Shanghai 201103, China. [6]Department of Nutrition and Food Hygiene, Hubei Key Laboratory of Food Nutrition and Safety, School of Public Health, Tongji Medical College, Huazhong University of Science and Technology, Wuhan 430030, China. [7]Department of Gastrointestinal Surgery, Zhongnan Hospital of Wuhan University, Wuhan University, Wuhan 430071, China. [8]Department of Psychiatry, Renmin Hospital of Wuhan University, Wuhan 430071, China. [9]TaiKang Center for Life and Medical Sciences, Wuhan University, Wuhan 430071, China. [10]Shanghai Artificial Intelligence Laboratory, Shanghai 200232, China. [11]These authors contributed equally: Caibo Ning, Linyun Fan, Meng Jin, Wenji Wang, Zhiqiang Hu, Yimin Cai, Liangkai Chen. ✉e-mail: Zhangshaoting@pjlab.org.cn; tianjb@whu.edu.cn; xpmiao@whu.edu.cn

