## [Peer Review File · Nature Communications]

Genome-wide association analysis of left ventricular imaging-derived phenotypes identifies 72 risk loci and yields genetic insights into hypertrophic cardiomyopathyREVIEWER COMMENTS

Reviewer #1 (Remarks to the Author):

This interesting study calculated 12 left ventricular regional wall thicknesses (LVRWTs) using a deep-learning framework named MSMM, and identified significant genetic loci associated with LVRWTs. These risk variants may influence disease risk via their regulatory activities, and candidate genes may contribute to the development of cardiomyopathies. A significant relationship was observed between the polygenic risk score of LVRWT and incident HCM. This study is well-designed and executed, yielding important insights into the genetic determinants of LVRWT phenotypes and shedding light on the biological basis for HCM etiology. However, the following issues, if addressed, might further support their conclusions.

Major issues

1. In this article, the authors propose a new artificial intelligence algorithm (MSMM) that accurately segments and measures the left ventricular wall, dividing it into six regions, each corresponding to two periods: end systole (ES) and end diastole (ED). Please explain further how MSMM algorithm is superior to other algorithms.
2. In lines 35-136 and in the section "Identification and functional annotation of susceptible genes associated with left ventricular regional wall thicknesses", the authors mention that there are variants and genes that affect multiple LVRWTs, and these pleiotropic variants and genes are very important. The authors should draw corresponding graphics (such as heat maps) to visually show the distribution of these pleiotropic variants and genes across multiple LVRWTs.
3. The authors used the MSMM artificial intelligence algorithm to segment and measure the left ventricular wall and obtained 12 LVRWT traits. However, there are few studies on such a fine division of wall thickness, especially in such a large sample population. In Figure 5A, the heritability range of these 12 wall thicknesses is 0.17-0.28. Are these values within the normal range? Are there other related studies that can support this?
4. The authors conducted two-sample Mendelian randomization analyses to further explore the causal relationship between LVRWT traits and cardiovascular diseases. Please add the GWAS summary of the instrumental variables used in MR analyses in the supplementary table.
5. The author constructed PRS in 12 LVRWT traits and explored the effect of PRS on HCM in 367,831 people without CMR, achieving good results. It is recommended that the authors conduct prediction in the population with CMR or in all UKB populations.

Minor issues

1. The word "variant" in line 175 should be used in plural form.

2. "HEART FAILURE" in line 483 should be changed to all lowercase. Please correct.
3. Please replace the term "nearestGene" in Supplementary Table 12 with "Nearest gene".
4. To avoid confusion with the p-values in the hazard ratio analyses, please label the p-values in Figures 7C and D, as well as Supplementary Figures 6-10C and D, as "log-rank P".

Reviewer #2 (Remarks to the Author):

The authors used a deep-learning model to quantify left ventricular regional wall thickness (LVRWT) measured by cardiac magnetic resonance (CMR) and identified 72 genetic loci associated with LVRWT phenotypes, which represents a substantial advance in the understanding of the genetic architecture of cardiovascular diseases. The observation of significant causal relationships between LVRWT traits and hypertrophic cardiomyopathy (HCM) using Mendelian randomization analyses provided valuable insights into the biological basis for HCM etiology. Furthermore, they found that PRSs derived from LVRWT traits predict incident HCM in individuals without CMR data, which has important implications for personalized risk stratification strategies. The use of deep learning algorithms to measure complex cardiovascular phenotypes is particularly noteworthy and has the potential to advance the knowledge of cardiovascular diseases (CVDs). Generally, the manuscript is well written, and the findings are novel and have important implications for CVD precision prevention. However, certain findings require further elaboration to substantiate the author's assertions before being published.

Major issues

1. When quantifying the myocardial thicknesses, the authors divided it into end systole and end diastole. What was the reason for dividing the LVRWT traits into systolic and diastolic phases, and is there any evidence in the results of the study to demonstrate that this division is better than not dividing it?
2. A recently published article (Khurshid et al, PMID: 36944631) is similar to this research. It conducted a GWAS of cardiac magnetic resonance-derived left ventricular mass indexed (LVMI) to body surface area among 43,230 participants in the UK Biobank and identified 12 genes associated with cardiac contractility and cardiomyopathy. Please compare these 12 genes with those found in this paper.
3. As previously mentioned, Khurshid et al.'s article is similar to your research in that it analyzes the genetic function of the left ventricular wall. However, the perspectives of analysis are different: Khurshid et al. measured the overall mass of the left ventricular wall, while your work started from the geometric angle of wall thickness. I am interested in whether there is a strong genetic correlation between LVMI and LVRWT, so please supplement your analysis on this part.
4. The authors performed genetic correlation analyses with GWAS summary statistics from 11 selected CVD traits. Could you explain why these specific 11 heart diseases were chosen?

5. From the overall procedure of the article, the authors first investigated the genetic correlation between LVRWT traits and heart disease, then conducted causal validation using two-sample Mendelian randomization, and finally performed PRS prediction for HCM. Could you explain why HCM was chosen for the PRS prediction?

6. The authors constructed PRS for 12 LVRWT traits and divided non-imaging populations into high-, medium-, and low-risk groups based on the tertiles of PRS. This division can indeed predict HCM occurrence well for some wall thicknesses, but for some ED-related wall thickness sources of PRS, it may cause indistinguishable medium- and low-risk groups (as shown in Supplementary Fig 8D). Therefore, it is suggested to divide the medium- and low-risk groups into subgroups for corresponding analysis.

7. Line 262: “remaining 367,831 individuals without CMR imaging data”. The description of the source of individuals without CMR imaging is unclear. Please provide a detailed explanation of the screening process for this population.

Minor issues

1. Line 243: “rage”, maybe replace by “range”.

2. The author only presented the r^2 of phenotype correlations in Supplementary Fig 2 without significant annotations. Please add annotations of P-values of phenotype correlations.

3. The authors performed Mendelian randomization analyses using two-sample methods to investigate the causal association between LVRWT traits and cardiovascular diseases. The GWAS summary information of MR analyses should be provided.

4. While the manuscript was understandable, the English writing could be improved with some refinement.

Reviewer #3 (Remarks to the Author):

Ning et al described the genetic architecture of left ventricular regional wall thickness (LVRWT) traits. The authors identified 72 genetic loci associated with LVRWT traits. They indicated that risk variants may regulate the expression of genes by activating chromatin states, which enhanced the understanding of the mechanisms underlying GWAS risk loci. Interestingly, the authors build polygenic risk scores (PRS) weighting the genetic dosage from each LVRWT trait GWAS, which was found to have significant relationships with incident hypertrophic cardiomyopathy (HCM). The paper is well-performed, and I have only a few comments regarding its acceptance.

Major issues

1. The authors mentioned in lines 401-402 that individuals with a history of previous myocardial infarction or a diagnosis of heart failure were excluded. Please explain the exclusion.

2. Protein modification includes various forms such as methylation, acetylation, phosphorylation, etc. They can affect the transcription, replication, and repair of genes by changing the structure and function of chromatin. It is an important regulatory mechanism in biological processes such as cell differentiation, development, and disease occurrence. In this article, the authors chose H3K4me1, H3K4me3, H3K27ac, H3K36me3 and H3K9me3 to examine the risk variants enrichment situation. There are hundreds of histone modifications available. Please explain in detail why the authors chose these histone modifications.

3. In Figure 3, the authors provided rich annotations for the obtained risk variants, including location annotations, histone enrichment analysis, and disease enrichment analysis. These annotations are meaningful, especially as Figure 3D shows that risk variants are significantly enriched in some heart-related diseases. Currently, single-cell-related research is booming, and the authors may consider examining whether these risk variants are specifically enriched in certain heart or myocardial-related cells, which may further explain the findings in Figure 3D.

4. The authors investigated the genetic correlation among 12 LVRWT traits in Figure 5A and found that most of the LVRWT traits are strongly correlated. However, in Figure 5B, the authors further explored the genetic correlation between these LVRWT traits and 10 traits related to cardiac structure and function. Why did the author take this step and not skip directly to the procedure in Figure 5C?

5. Cardiomyopathy was included as one of the 11 cardiovascular diseases analyzed in the genetic correlation study of left ventricular wall thickness measures, as described in lines 482-484. However, the authors specifically chose to investigate dilated cardiomyopathy and hypertrophic cardiomyopathy as distinct subtypes. Please provide a detailed explanation for why selected these two cardiomyopathy subtypes for analysis, despite inclusion under the broader category of cardiomyopathy?

6. The authors constructed PRS for 12 LVRWT traits and predicted the incidence of HCM in non-CMR populations, and found that some of the PRS for LVRWT were predictive of the occurrence of HCM. However, it is unclear whether these PRS predict other heart diseases, such as dilated cardiomyopathy.

Minor issues

1. The abbreviation 'EA' in the title row of Table 1 spans two lines, please make corrections.

2. It is better to label the p-values in Figure 7CD as "log-rank P".

3. Minor Allele Frequency (MAF) can be included as an additional column in Supplementary Table 18.

The manuscript comprises a vast array of diverse information. The author should enhance the coherence of the text.

Reviewer #4 (Remarks to the Author):

The authors performed a GWAS on left ventricular regional wall thickness LVRWT, followed by a substantial number of post-GWAS analyses. By and large while the methods are fairly robust (and standard, which is a good thing), the manuscript lacks important details on outcome definitions and data sources.

Major comments:

1. The authors presumably use aggregated GWAS data for their the genetic correlation and the Mendelian randomization analyses. The source of these data is however not provided at all in their manuscript, or at least not in the data availability section, the methods or the figure legends. Without this information the reader is left wondering how these estimate magically appear, and above all cannot attempt to replicate the presented findings.
2. Similarly, but distinct, for the PRS analysis the authors attempt to predict HCM onset in UKB participants without CMR data. Great idea of course. However, the authors do not explain how HCM was defined, whether this is a HCM diagnosis (I hope) or merely carriership? Do the HCM cases pertain to incidence (after UKB enrolment), or also include prevalent cases? Given that PRS is about prediction, including prevalent cases is of course irrelevant. The major problem of course is that I do not have any idea if this comment on prevalent or incidence is important here, the authors have not provided any explanation. At the minimal please include a code list how HCM was defined and at what time relevant to UKB enrolment.
3. On the presentation of the PRS, while I understand that the presented scores significantly associate with HCM (however defined), and the author do provide HR for low, intermediate and high PRS groups, this presentation has been shown to be insufficient in the statistical literature for decades; see Royston BMC MRM 2013. Please, at least additional (if not substitute) provide measures of discrimination such as the c-statistics or AUC. Based on the presented results I am confident the c-statistic/AUC will be extremely close to 0.50 suggesting the benefit of these PRS are likely minimal. In all fairness, most genetic PRS studies to present their results without c-stat/AUC, presumably because these are simply fairly small and unlike the HR not bounded between 0 and 1.
4. Have the authors considered combining multiple trait specific PRS into a stacked model to predict HCM. See <https://www.medrxiv.org/content/10.1101/2022.09.01.22279477v1>
5. To what extend are the LVRWT measurements correlated? Could the authors please include a principle component analysis and identify how many PCs were necessary to explain 90% of the variability in the LVRWT measurements. This would give an indication of the multiple testing threshold.
6. Could the authors please identify which signals passed the canonical GWAS significance threshold corrected for the number fo PCs (see above).
7. Regarding the MR analyses and the multiple testing correction it seems the authors adjusted the p-value threshold for the number of methods used (5). Given that these methods all use the same (or

subset of the same) data, and are therefore correlated, such a correction is inappropriate. Instead please correct for the number of outcomes (11) times the number of PCs necessary to explain 90% of the variability in the CMR traits.

8. The authors do not seem to be planning on sharing their aggregated GWAS results (based on the data availability section). This would be extremely poor conduct, especially because most of their analyses just discussed used other peoples shared GWAS results. The authors should immediately and fully share all their aggregate results, which do not have any privacy concerns. Please do not share them via some weird online shiny app or otherwise “data friendly” website – which is a form of data obfuscation , and simply share the gzipped files, include the effect allele, beta coefficients, standard error and p-values. Here is the GWAS catalog deposition link: <https://www.ebi.ac.uk/gwas/deposition>

9. The author seem to have overlooked a few recent GWAS on CMR trait which are relevant here: <https://pubmed.ncbi.nlm.nih.gov/36598836/>
<https://www.science.org/doi/10.1126/sciadv.add4984>

Reviewer #5 (Remarks to the Author):

1. It is a big leap of faith in saying that a model trained on ACDC dataset is good enough to segment the UKB population data. Especially when the authors mention the use of pathological and healthy images for ACDC and UKB is largely a healthy population. And that is why it is hard to verify the accuracy of the proposed DLA model. Authors should put effort in addressing this issue and help in bridging this gap.

Some notes:

a. The details of the deep learning architecture are quite poorly explained by the authors even in the supplementary notes. It would be useful to include the more details like number of hidden layers, some example segmentations that show accuracy of the models.

b. The testing dataset is quite small for the DLA model, I wonder how sensitive the LV wall thickness calculations are to the number of images as input for DLA architecture training.

c. Please include the time-performance of the DLA method on the said GPU you have used for segmentation.

d. The following sentence seems a bit unscientific: “As we expected, after a visual evaluation of segmentation results on random 100 images by an experienced doctor, it could be considered that the model also performed well on the UKB dataset.”

i. I think a more scientific approach is required to quantify the performance. For example, the clinicians can blindly segment the images and then you could use some sort of score to compare the segmentations. I think the authors are assuming that clinical segmentations are ground truth, which is understandable so a direct comparison of segmentations of untrained images is warranted.

Responses to the reviewers' comments

Reviewer #1

This interesting study calculated 12 left ventricular regional wall thicknesses (LVRWTs) using a deep-learning framework named MSMM, and identified significant genetic loci associated with LVRWTs. These risk variants may influence disease risk via their regulatory activities, and candidate genes may contribute to the development of cardiomyopathies. A significant relationship was observed between the polygenic risk score of LVRWT and incident HCM. This study is well-designed and executed, yielding important insights into the genetic determinants of LVRWT phenotypes and shedding light on the biological basis for HCM etiology. However, the following issues, if addressed, might further support their conclusions.

Response: Thanks for the positive comments. We have addressed the concerns in the following responses.

Major issues

1. In this article, the authors propose a new artificial intelligence algorithm (MSMM) that accurately segments and measures the left ventricular wall, dividing it into six regions, each corresponding to two periods: end systole (ES) and end diastole (ED). Please explain further how MSMM algorithm is superior to other algorithms.

Response: Thanks for your comments. The proposed MSMM consists of two components: one for myocardial segmentation and another for automatic quantification of myocardial wall thicknesses. For segmentation, we employed the DLANet, a model that utilizes Hierarchical Deep Aggregation (HDA) and Iterative Deep Aggregation (IDA) to merge information from multiple layers and scales. With the HDA and IDA, the DLANet can better merge information from multiple layers and scales, which can extract more robust features and output accurate results.

To evaluate the effectiveness of DLANet, we compared it with state-of-the-art methods by re-implementing them and reporting the results on different datasets in Supplementary Table 22 and Supplementary Table 23. Since only the mid-cavity slices' quantification results were used for the subsequent analysis, we evaluated the methods on selected slices of each case to ensure a consistent comparison. In Supplementary Table 22, DLANet achieved the best segmentation results in terms of both Dice and HD metrics. Additionally, for the UKB dataset, all testing results surpassed those of the ACDC dataset due to the majority of UKB cases being healthy, making the task of segmentation easier. Supplementary Table 23 shows that DLANet also achieved a good overall performance

with an average Dice of 0.962 and an average HD of 2.051 mm.

During the quantification stage, we utilized the mean absolute error (MAE) metric to evaluate the different methods. All methods achieved accurate quantification results, with the MAE of LVRWT are at most of 0.987 mm for the ACDC dataset and 0.745 mm for the UKB dataset. We would like to emphasize that these values are comparable to the spacing represented by one pixel (ranging from 0.70 mm/pixel to 1.92 mm/pixel), and as a result demonstrate the models' reliability on clinical practice. The MSMM achieved the best MAE of 0.853 ± 0.304 for ACDC and the second-best MAE of 0.663 ± 0.343 for UKB.

In summary, all experimental results demonstrate that the MSMM outperforms other algorithms, including very widely used methods like TransUNet. Moreover, it already fulfills the requirements of practical applications. We have added the details in the revised manuscript and Supplementary Methods.

Supplementary Table 22. Testing results on ACDC dataset with different methods for comparison.

Methods	Dice (\uparrow)		HD (mm, \downarrow)		MAE (mm, \downarrow)
	LVC	LVM	LVC	LVM	
VNet	0.967 ± 0.022	0.913 ± 0.030	2.906 ± 1.829	3.502 ± 2.046	0.887 ± 0.348
TransUNet w/o pretrain	0.965 ± 0.028	0.909 ± 0.026	2.704 ± 1.160	3.448 ± 1.327	0.869 ± 0.408
TransUNet w pretrain	0.964 ± 0.024	0.904 ± 0.029	3.030 ± 1.303	3.640 ± 1.336	0.987 ± 0.358
DLANet	0.968 ± 0.018	0.915 ± 0.025	2.545 ± 0.787	3.171 ± 0.935	0.853 ± 0.304

Methods include VNet, TransUNet without pretrained ViT, TransUNet with pretrained ViT and DLANet. The best results are highlighted in bold. Aberrations: Dice: Dice coefficient, HD: Hausdorff Distance, MAE: mean absolute errors. LVC, left ventricular cavity; LVM, left ventricular myocardium

Supplementary Table 23. Testing results on UKB dataset with different methods for comparison.

Methods	Dice (\uparrow)		HD (mm, \downarrow)		MAE (mm, \downarrow)
	LVC	LVM	LVC	LVM	
VNet	0.975 ± 0.020	0.948 ± 0.021	1.865 ± 0.981	2.363 ± 1.075	0.675 ± 0.351
TransUNet w/o pretrain	0.975 ± 0.026	0.948 ± 0.025	1.892 ± 1.144	2.350 ± 1.374	0.637 ± 0.338

| | | | | | |
|---|---|---|---|---|---|
| **TransUNet w pretrain** | 0.973±0.022 | 0.943±0.024 | 1.987±1.041 | 2.514±1.155 | 0.745±0.409 |
| **DLANet** | 0.974±0.021 | **0.949±0.020** | **1.865±0.932** | **2.236±1.014** | 0.663±0.343 |

Models are trained on ACDC dataset. The best results are highlighted in bold. Aberrations: Dice: Dice coefficient, HD: Hausdorff Distance, MAE: mean absolute errors. LVC, left ventricular cavity; LVM, left ventricular myocardium

2. In lines 35-136 and in the section “Identification and functional annotation of susceptible genes associated with left ventricular regional wall thicknesses”, the authors mention that there are variants and genes that affect multiple LVRWTs, and these pleiotropic variants and genes are very important. The authors should draw corresponding graphics (such as heat maps) to visually show the distribution of these pleiotropic variants and genes across multiple LVRWTs.

**Response**: Thank you for your valuable suggestion. We have created corresponding heat maps, to depict the distribution of pleiotropic variants and genes across multiple LVRWTs in the revised Supplementary Fig. 5 and Supplementary Fig. 7.

**In line 150-155, we now state:**

Interestingly, there have 10 genetic variants associated with multiple LVRWT traits, resulting in 62 unique variants associated with the 12 LVRWT traits; for example, rs7928419 in *SPI1* at 11p11.2 reached genome-wide significance in ES-IS, ED-IS and ED-AS ($P_{ES\text{-}IS} = 1.0\times10^{-12}$, $P_{ED\text{-}IS} = 1.3\times10^{-9}$, $P_{ED\text{-}AS} = 1.2\times10^{-8}$, **Supplementary Fig. 5**)

Supplementary Fig. 5. Heatmap of pleiotropic variants distribution of each LVRWTs

Abbreviation: LVRWT, LV regional wall thickness; IS, inferoseptal; I, inferior; IL, inferolateral; AL, anterolateral; A, anterior; AS, anterospetal; ES, end systole; ED, end diastole.

In line 194-196, we now state:

ALPK3, *NMB*, and *WNT3* loci, which are significant candidate genes that have been linked to inherited CVDs, were shared across most LVRWT traits (**Supplementary Fig. 7**).

Supplementary Fig. 7. Heatmap of pleiotropic gene distribution of each LVRWTs using four methods

The times of the gene annotated by four methods (Nearest Gene, eQTL, MAGMA, TWAS) are indicated by the color legend. For example, the *ALPK3* gene can be annotated by the four methods in end-diastole inferior LVRWT. The maximum number of each cell is 4. Further details can see in Supplementary Table 12. Abbreviation: LVRWT, LV regional wall thickness; IS, inferoseptal; I, inferior; IL, inferolateral; AL, anterolateral; A, anterior; AS, anterospetal; ES, end systole; ED, end diastole.

3. The authors used the MSMM artificial intelligence algorithm to segment and measure the left ventricular wall and obtained 12 LVRWT traits. However, there are few studies on such a fine division of wall thickness, especially in such a large sample population. In Figure 5A, the heritability range of these 12 wall thicknesses is 0.17-0.28. Are these values within the normal range? Are there other related studies that can support this?

Response: Thank you for your question. The heritabilities of the 12 LVRWT traits are comparable to existing research, such as LV volume and LV ejection fraction, which have shown 22% to 39% heritability (Aung et al. *Circulation*. 2019;140(16):1318-1330). This consistency implies that the observed heritability values for the 12 LVRWT traits fall within the expected range for LV-related characteristics.

4. The authors conducted two-sample Mendelian randomization analyses to further explore the causal relationship between LVRWT traits and cardiovascular diseases. Please add the GWAS summary of the instrumental variables used in MR analyses in the supplementary table.

Response: Thank you for the comment. We have incorporated the GWAS summary of the instrumental variables utilized in the Mendelian randomization analyses in Supplementary Table 3.

5. The author constructed PRS in 12 LVRWT traits and explored the effect of PRS on HCM in 367,831 people without CMR, achieving good results. It is recommended that the authors conduct prediction in the population with CMR or in all UKB populations.

Response: Thank you for the comment. We have conducted additional analyses in both the population with CMR and the entire UKB population as you recommended. The results are summarized in the provided table. Notably, the outcomes for "without CMR" and "all UKB populations" were consistent (Table R1 and R2). Specifically, the analysis demonstrated that the PRS tertiles of inferoseptal LVRWT at end systole were consistently associated with a higher risk of HCM across both of these population subsets, with HR (95% CI) of 1.69 (95% CI: 1.33-2.15) for "without CMR" and 1.64 (95% CI: 1.29-2.07) for "all UKB populations". However, the analysis for "with CMR" rendering the findings less meaningful, due to the limited sample size (Table R3). After thorough consideration of various factors, we decided to retain the previous results in the paper.

Table R1. Associations between LVRWT PRS and incident hypertrophic cardiomyopathy in the population without CMR.

Period	Trait	Disease	Follow-up, yrs (Q1-Q3)	N events/N total	PRS tertiles	P (log-rank)	P (HR)	HR (95%CI)
ES	IS	Hypertrophic cardiomyopathy	11.82 (11.10-12.53)	420/439981	High	8.12e-07	2.31e-05	1.69 (95% CI: 1.33-2.15)
ES	IS	Hypertrophic cardiomyopathy	11.82 (11.10-12.53)	420/439981	Intermediate	8.12e-07	3.09e-01	1.14 (95% CI: 0.88-1.48)
ES	I	Hypertrophic cardiomyopathy	11.82 (11.10-12.53)	420/439981	High	1.10e-01	1.52e-01	1.19 (95% CI: 0.94-1.51)
ES	I	Hypertrophic cardiomyopathy	11.82 (11.10-12.53)	420/439981	Intermediate	1.10e-01	5.77e-01	1.07 (95% CI: 0.84-1.37)
ES	IL	Hypertrophic cardiomyopathy	11.82 (11.10-12.53)	420/439981	High	3.86e-06	1.57e-05	1.67 (95% CI: 1.32-2.10)
ES	IL	Hypertrophic cardiomyopathy	11.82 (11.10-12.53)	420/439981	Intermediate	3.86e-06	6.46e-01	0.94 (95% CI: 0.73-1.22)
ES	AL	Hypertrophic cardiomyopathy	11.82 (11.10-12.53)	420/439981	High	3.62e-06	5.56e-05	1.63 (95% CI: 1.29-2.07)
ES	AL	Hypertrophic cardiomyopathy	11.82 (11.10-12.53)	420/439981	Intermediate	3.62e-06	5.95e-01	1.07 (95% CI: 0.83-1.39)

ES	A	Hypertrophic cardiomyopathy	11.82 (11.10-12.53)	420/439981	High	4.24e-03	8.43e-03	1.38 (95% CI: 1.09-1.75)
ES	A	Hypertrophic cardiomyopathy	11.82 (11.10-12.53)	420/439981	Intermediate	4.24e-03	5.08e-01	1.09 (95% CI: 0.85-1.40)
ES	AS	Hypertrophic cardiomyopathy	11.82 (11.10-12.53)	420/439981	High	1.31e-05	1.76e-04	1.58 (95% CI: 1.25-2.01)
ES	AS	Hypertrophic cardiomyopathy	11.82 (11.10-12.53)	420/439981	Intermediate	1.31e-05	4.54e-01	1.10 (95% CI: 0.85-1.43)
ED	IS	Hypertrophic cardiomyopathy	11.82 (11.10-12.53)	420/439981	High	8.71e-01	4.62e-01	1.09 (95% CI: 0.86-1.39)
ED	IS	Hypertrophic cardiomyopathy	11.82 (11.10-12.53)	420/439981	Intermediate	8.71e-01	6.14e-01	1.06 (95% CI: 0.84-1.35)
ED	I	Hypertrophic cardiomyopathy	11.82 (11.10-12.53)	420/439981	High	4.87e-04	3.95e-04	1.55 (95% CI: 1.22-1.98)
ED	I	Hypertrophic cardiomyopathy	11.82 (11.10-12.53)	420/439981	Intermediate	4.87e-04	3.12e-02	1.32 (95% CI: 1.03-1.70)
ED	IL	Hypertrophic cardiomyopathy	11.82 (11.10-12.53)	420/439981	High	2.92e-03	1.69e-03	1.49 (95% CI: 1.16-1.92)
ED	IL	Hypertrophic cardiomyopathy	11.82 (11.10-12.53)	420/439981	Intermediate	2.92e-03	4.40e-04	1.55 (95% CI: 1.21-1.98)
ED	AL	Hypertrophic cardiomyopathy	11.82 (11.10-12.53)	420/439981	High	2.14e-01	1.87e-02	1.34 (95% CI: 1.05-1.70)
ED	AL	Hypertrophic cardiomyopathy	11.82 (11.10-12.53)	420/439981	Intermediate	2.14e-01	5.43e-02	1.27 (95% CI: 1.00-1.61)
ED	A	Hypertrophic cardiomyopathy	11.82 (11.10-12.53)	420/439981	High	4.37e-03	1.55e-02	1.33 (95% CI: 1.06-1.67)
ED	A	Hypertrophic cardiomyopathy	11.82 (11.10-12.53)	420/439981	Intermediate	4.37e-03	2.57e-01	0.87 (95% CI: 0.67-1.11)
ED	AS	Hypertrophic cardiomyopathy	11.82 (11.10-12.53)	420/439981	High	4.51e-01	3.19e-01	1.13 (95% CI: 0.89-1.42)
ED	AS	Hypertrophic cardiomyopathy	11.82 (11.10-12.53)	420/439981	Intermediate	4.51e-01	7.09e-01	0.96 (95% CI: 0.75-1.22)

Table R2. Associations between LVRWT PRS and incident hypertrophic cardiomyopathy in all UKB populations.

Period	Trait	Disease	Follow-up, yrs (Q1-Q3)	N events/N total	PRS tertiles	P (log-rank)	P (HR)	HR (95%CI)
ES	IS	Hypertrophic cardiomyopathy	11.84 (11.12-12.54)	444/484065	High	1.18e-06	3.79e-05	1.64 (95% CI: 1.29-2.07)
ES	IS	Hypertrophic cardiomyopathy	11.84 (11.12-12.54)	444/484065	Intermediate	1.18e-06	4.27e-01	1.11 (95% CI: 0.86-1.42)
ES	I	Hypertrophic cardiomyopathy	11.84 (11.12-12.54)	444/484065	High	8.78e-02	1.30e-01	1.20 (95% CI: 0.95-1.51)
ES	I	Hypertrophic cardiomyopathy	11.84 (11.12-12.54)	444/484065	Intermediate	8.78e-02	6.63e-01	1.05 (95% CI: 0.83-1.34)
ES	IL	Hypertrophic cardiomyopathy	11.84 (11.12-12.54)	444/484065	High	4.16e-06	2.43e-05	1.62 (95% CI: 1.30-2.03)
ES	IL	Hypertrophic cardiomyopathy	11.84 (11.12-12.54)	444/484065	Intermediate	4.16e-06	5.24e-01	0.92 (95% CI: 0.72-1.18)
ES	AL	Hypertrophic cardiomyopathy	11.84 (11.12-12.54)	444/484065	High	3.95e-06	3.88e-05	1.63 (95% CI: 1.29-2.05)
ES	AL	Hypertrophic cardiomyopathy	11.84 (11.12-12.54)	444/484065	Intermediate	3.95e-06	4.76e-01	1.10 (95% CI: 0.85-1.41)
ES	A	Hypertrophic cardiomyopathy	11.84 (11.12-12.54)	444/484065	High	4.29e-03	8.78e-03	1.36 (95% CI: 1.08-1.72)
ES	A	Hypertrophic cardiomyopathy	11.84 (11.12-12.54)	444/484065	Intermediate	4.29e-03	5.62e-01	1.07 (95% CI: 0.84-1.37)
ES	AS	Hypertrophic cardiomyopathy	11.84 (11.12-12.54)	444/484065	High	9.35e-06	1.32e-04	1.58 (95% CI: 1.25-1.99)
ES	AS	Hypertrophic cardiomyopathy	11.84 (11.12-12.54)	444/484065	Intermediate	9.35e-06	4.70e-01	1.10 (95% CI: 0.85-1.41)
ED	IS	Hypertrophic cardiomyopathy	11.84 (11.12-12.54)	444/484065	High	7.21e-01	3.14e-01	1.13 (95% CI: 0.89-1.42)
ED	IS	Hypertrophic cardiomyopathy	11.84 (11.12-12.54)	444/484065	Intermediate	7.21e-01	4.64e-01	1.09 (95% CI: 0.87-1.37)
ED	I	Hypertrophic cardiomyopathy	11.84 (11.12-12.54)	444/484065	High	1.69e-04	1.57e-04	1.58 (95% CI: 1.25-2.01)
ED	I	Hypertrophic cardiomyopathy	11.84 (11.12-12.54)	444/484065	Intermediate	1.69e-04	1.42e-02	1.36 (95% CI: 1.06-1.74)
ED	IL	Hypertrophic cardiomyopathy	11.84 (11.12-12.54)	444/484065	High	2.57e-03	7.00e-04	1.52 (95% CI: 1.19-1.94)
ED	IL	Hypertrophic cardiomyopathy	11.84 (11.12-12.54)	444/484065	Intermediate	2.57e-03	5.97e-04	1.52 (95% CI: 1.20-1.93)
ED	AL	Hypertrophic cardiomyopathy	11.84 (11.12-12.54)	444/484065	High	2.29e-01	1.90e-02	1.32 (95% CI: 1.05-1.68)
ED	AL	Hypertrophic cardiomyopathy	11.84 (11.12-12.54)	444/484065	Intermediate	2.29e-01	6.81e-02	1.24 (95% CI: 0.98-1.57)
ED	A	Hypertrophic cardiomyopathy	11.84 (11.12-12.54)	444/484065	High	6.81e-03	1.39e-02	1.32 (95% CI: 1.06-1.65)
ED	A	Hypertrophic cardiomyopathy	11.84 (11.12-12.54)	444/484065	Intermediate	6.81e-03	3.97e-01	0.90 (95% CI: 0.71-1.15)
ED	AS	Hypertrophic cardiomyopathy	11.84 (11.12-12.54)	444/484065	High	2.22e-01	1.77e-01	1.17 (95% CI: 0.93-1.47)
ED	AS	Hypertrophic cardiomyopathy	11.84 (11.12-12.54)	444/484065	Intermediate	2.22e-01	6.66e-01	0.95 (95% CI: 0.75-1.20)

Table R3. Associations between LVRWT PRS and incident hypertrophic cardiomyopathy in the population with CMR.

Period	Trait	Disease	Follow-up, yrs (Q1-Q3)	N events/N total	PRS tertiles	P (log-rank)	P (HR)	HR (95%CI)
ES	IS	Hypertrophic cardiomyopathy	11.98 (11.24-12.62)	24/44084	High	6.84e-01	9.17e-01	1.05 (95% CI: 0.42-2.65)
ES	IS	Hypertrophic cardiomyopathy	11.98 (11.24-12.62)	24/44084	Intermediate	6.84e-01	4.39e-01	0.66 (95% CI: 0.24-1.87)
ES	I	Hypertrophic cardiomyopathy	11.98 (11.24-12.62)	24/44084	High	6.06e-01	5.98e-01	1.28 (95% CI: 0.51-3.26)
ES	I	Hypertrophic cardiomyopathy	11.98 (11.24-12.62)	24/44084	Intermediate	6.06e-01	5.93e-01	0.75 (95% CI: 0.26-2.16)
ES	IL	Hypertrophic cardiomyopathy	11.98 (11.24-12.62)	24/44084	High	6.85e-01	8.84e-01	1.07 (95% CI: 0.42-2.71)
ES	IL	Hypertrophic cardiomyopathy	11.98 (11.24-12.62)	24/44084	Intermediate	6.85e-01	4.89e-01	0.69 (95% CI: 0.25-1.95)
ES	AL	Hypertrophic cardiomyopathy	11.98 (11.24-12.62)	24/44084	High	8.84e-01	5.19e-01	1.38 (95% CI: 0.51-3.72)
ES	AL	Hypertrophic cardiomyopathy	11.98 (11.24-12.62)	24/44084	Intermediate	8.84e-01	7.69e-01	1.16 (95% CI: 0.42-3.21)
ES	A	Hypertrophic cardiomyopathy	11.98 (11.24-12.62)	24/44084	High	8.82e-01	5.51e-01	1.35 (95% CI: 0.50-3.65)
ES	A	Hypertrophic cardiomyopathy	11.98 (11.24-12.62)	24/44084	Intermediate	8.82e-01	7.33e-01	1.19 (95% CI: 0.43-3.30)
ES	AS	Hypertrophic cardiomyopathy	11.98 (11.24-12.62)	24/44084	High	6.85e-01	4.13e-01	1.50 (95% CI: 0.57-3.94)
ES	AS	Hypertrophic cardiomyopathy	11.98 (11.24-12.62)	24/44084	Intermediate	6.85e-01	9.62e-01	1.03 (95% CI: 0.36-2.93)
ED	IS	Hypertrophic cardiomyopathy	11.98 (11.24-12.62)	24/44084	High	4.13e-01	2.07e-01	2.00 (95% CI: 0.68-5.85)
ED	IS	Hypertrophic cardiomyopathy	11.98 (11.24-12.62)	24/44084	Intermediate	4.13e-01	2.64e-01	1.87 (95% CI: 0.62-5.57)
ED	I	Hypertrophic cardiomyopathy	11.98 (11.24-12.62)	24/44084	High	2.23e-01	1.29e-01	2.47 (95% CI: 0.77-7.90)
ED	I	Hypertrophic cardiomyopathy	11.98 (11.24-12.62)	24/44084	Intermediate	2.23e-01	1.14e-01	2.55 (95% CI: 0.80-8.14)
ED	IL	Hypertrophic cardiomyopathy	11.98 (11.24-12.62)	24/44084	High	9.21e-02	1.30e-01	2.11 (95% CI: 0.80-5.57)
ED	IL	Hypertrophic cardiomyopathy	11.98 (11.24-12.62)	24/44084	Intermediate	9.21e-02	7.52e-01	0.83 (95% CI: 0.25-2.71)
ED	AL	Hypertrophic cardiomyopathy	11.98 (11.24-12.62)	24/44084	High	8.83e-01	8.23e-01	1.11 (95% CI: 0.43-2.89)
ED	AL	Hypertrophic cardiomyopathy	11.98 (11.24-12.62)	24/44084	Intermediate	8.83e-01	8.00e-01	0.88 (95% CI: 0.32-2.42)
ED	A	Hypertrophic cardiomyopathy	11.98 (11.24-12.62)	24/44084	High	4.11e-01	8.33e-01	1.12 (95% CI: 0.38-3.35)
ED	A	Hypertrophic cardiomyopathy	11.98 (11.24-12.62)	24/44084	Intermediate	4.11e-01	2.08e-01	1.90 (95% CI: 0.70-5.13)
ED	AS	Hypertrophic cardiomyopathy	11.98 (11.24-12.62)	24/44084	High	9.17e-02	1.37e-01	2.09 (95% CI: 0.79-5.50)
ED	AS	Hypertrophic cardiomyopathy	11.98 (11.24-12.62)	24/44084	Intermediate	9.17e-02	7.26e-01	0.81 (95% CI: 0.25-2.65)

Minor issues

1. The word "variant" in line 175 should be used in plural form.

Response: Thank you for the comment. We have revised it in the revised manuscript.

In line 191-192, we now state:

Notably, *ALPK3* was annotated by the four methods, and rs3803405 in proximity to *ALPK3* was the most significant variants in this study.

2. "HEART FAILURE" in line 483 should be changed to all lowercase. Please correct.

Response: Thank you for the comment. We have revised it in the revised manuscript.

In line 524-526, we now state:

hypertension, pulmonary hypertension, dilated cardiomyopathy, HCM, cardiomyopathy, heart failure, atrial fibrillation, myocardial infarction, angina pectoris, chronic ischaemic heart disease, ischaemic stroke.

3. Please replace the term "nearestGene" in Supplementary Table 12 with "Nearest gene".

Response: Thank you for the comment. We have revised it in the revised Supplementary Table 12.

4. To avoid confusion with the p-values in the hazard ratio analyses, please label the p-values in Figures 7C and D, as well as Supplementary Figures 6-10C and D, as "log-rank P".

Response: Thank you for the comment. We have revised them in the revised Figures and Supplementary Figures.

Reviewer #2:

The authors used a deep-learning model to quantify left ventricular regional wall thickness (LVRWT) measured by cardiac magnetic resonance (CMR) and identified 72 genetic loci associated with LVRWT phenotypes, which represents a substantial advance in the understanding of the genetic architecture of cardiovascular diseases. The observation of significant causal relationships between LVRWT traits and hypertrophic cardiomyopathy (HCM) using Mendelian randomization analyses provided valuable insights into the biological basis for HCM etiology. Furthermore, they found that PRSs derived from LVRWT traits predict incident HCM in individuals without CMR data, which has important implications for personalized risk stratification strategies. The use of deep learning algorithms to measure complex cardiovascular phenotypes is particularly noteworthy and has the potential to advance the knowledge of cardiovascular diseases (CVDs). Generally, the manuscript is well written, and the findings are novel and have important implications for CVD precision prevention. However, certain findings require further elaboration to substantiate the author's assertions before being published.

Response: Thank you for your insightful comments. We have addressed the concerns in the following responses.

Major issues

1. When quantifying the myocardial thicknesses, the authors divided it into end systole and end diastole. What was the reason for dividing the LVRWT traits into systolic and diastolic phases, and is there any evidence in the results of the study to demonstrate that this division is better than not dividing it?

Response: Thank you for your question. The division of LVRWT traits into systolic and diastolic phases aims to capture potential genetic associations and physiological differences between these distinct cardiac phases. It's worth noting that several other studies have also employed similar strategies (Pérez-Pelegri et al. *Comput Med Imaging Graph.* 2022;99:102085), dividing the LVRWT traits into systolic and diastolic phases. Our findings indicated significant genetic associations and heritabilities for LVRWT traits in both phases, suggesting that exploring them separately could provide valuable insights into genetic determinants of left ventricular remodeling dynamics.

2. A recently published article (Khurshid et al, PMID: 36944631) is similar to this research. It conducted a GWAS of cardiac magnetic resonance-derived left ventricular mass indexed (LVMI) to body surface area among 43,230 participants in the UK Biobank and identified 12 genes associated with cardiac contractility and cardiomyopathy. Please compare these 12 genes with those found in this paper.

Response: Thank you for the comment. We have compared the genes identified in Khurshid's article with our findings. Upon examining the loci of the 13 genes reported in their study and annotating the nearest genes to those loci, we observed an overlap with 5 genes that were also identified in our research (Table R1). This overlap highlights the consistency of genetic associations related to cardiac traits and cardiomyopathy between our study and Khurshid's study. We have added the details in the revised manuscript.

In line 338-343, we now state:

In light of recent Khurshid's study, it conducted a GWAS of CMR-derived left ventricular mass indexed (LVMI) in the UKB. By comparing the loci of the 13 genes from Khurshid's study, we found an overlap with 5 genes also identified in our research. This overlap underscores the consistency of genetic associations related to cardiac traits and cardiomyopathy between our study and Khurshid's study.

Table R1. Comparison of commonly identified genes between the current study and Khurshid et al.

Khurshid gene	Chr	Start	End	Loci	Nearest gene in our paper
CLCN6	1	11866207	11903201	1p36.22	CLCN6, MTHFR
TTN	2	179390716	179695529	2q31.2	-
CCDC141	2	179694484	179914813	2q31.2	-
HSPA4	5	132387654	132442141	5q31.1	HSPA4
CENPW	6	126661320	126670021	6q22.32	-
SYNPO2L	10	75404639	75423561	10q22.2	SYNPO2L
MYBPC3	11	47352957	47374253	11p11.2	SPI1
KDM2B	12	121866900	122018920	12q24.31	-
IGF1R	15	99192200	99507759	15q26.3	IGF1R
PDXDC1	16	15068448	15233196	16p13.11	-
MAPT	17	43971748	44105700	17q21.31	ARL17B, LINC02210-CRHR1, MAPT, MAPT-AS1
WNT3	17	44839872	44910520	17q21.32	RPRML, GOSR2
ADAMTS10	19	8645126	8675619	19p13.2	-

3. As previously mentioned, Khurshid et al.'s article is similar to your research in that it analyzes the genetic function of the left ventricular wall. However, the perspectives of analysis are different: Khurshid et al. measured the overall mass of the left ventricular wall, while your work started from the geometric angle of wall thickness. I am interested in whether there is a strong genetic correlation between LVMI and LVRWT, so please supplement your analysis on this part.

Response: Thank you for the comment. We have thoroughly explored the genetic correlation between LVMI and LVRWT. Our findings suggest a notable genetic correlation between these two cardiac traits (Figure R1).

Figure R1. Genetic correlation analysis between LVRWT and LVMI

4. The authors performed genetic correlation analyses with GWAS summary statistics from 11 selected CVD traits. Could you explain why these specific 11 heart diseases were chosen?

Response: Thank you for your question. The 11 CVDs selected for genetic correlation analyses were chosen based on their prevalence, clinical significance, and relevance to the field of cardiovascular research (Roth et al. *J Am Coll Cardiol.* 2020;76(25):2982-3021; Ahlberg et al. *Eur Heart J.* 2021; 42(44):4523-4534; Aung et al. *Nat Genet.* 2022;54(6):783-791). These CVDs represent a diverse range of common and impactful cardiovascular conditions, allowing our study to explore potential shared genetic factors and pathways that contribute to their underlying mechanisms.

5. From the overall procedure of the article, the authors first investigated the genetic correlation between LVRWT traits and heart disease, then conducted causal validation using two-sample Mendelian randomization, and finally performed PRS prediction for HCM. Could you explain why HCM was chosen for the PRS prediction?

Response: Thank you for your question. The selection of HCM for PRS prediction was driven by several key factors. Firstly, we observed that HCM exhibited the highest genetic correlation estimates with LVRWT among the investigated cardiovascular diseases. Additionally, the selection of HCM for PRS prediction was informed by its clinical significance. Notably, HCM shares a clear physiological association with the LVRWT traits (Gersh et al. *Circulation.* 2011;124(24):2761-96).

6. The authors constructed PRS for 12 LVRWT traits and divided non-imaging populations into high-, medium-, and low-risk groups based on the tertiles of PRS. This

division can indeed predict HCM occurrence well for some wall thicknesses, but for some ED-related wall thickness sources of PRS, it may cause indistinguishable medium- and low-risk groups (as shown in Supplementary Fig 8D). Therefore, it is suggested to divide the medium- and low-risk groups into subgroups for corresponding analysis.

Response: Thank you for the comment. We conducted further analysis by dividing the medium- and low-risk groups into subgroups. Our results revealed that this subdivision did yield some differentiation between the subgroups, as anticipated (Table R2). Therefore, after careful consideration, we decided to maintain the original high-, medium-, and low-risk groups. This approach strikes a balance between complexity and interpretability while still providing meaningful insights into the PRS-based prediction of HCM.

Table R2. Associations between LVRWT PRS and incident hypertrophic cardiomyopathy in the population with CMR

Period	Trait	Disease	Follow-up, yrs (Q1-Q3)	N events/N total	PRS tertiles	P (log-rank)	P (HR)	HR (95%CI)
ES	IS	Hypertrophic cardiomyopathy	11.82 (11.10-12.53)	420/439981	High	1.93e-07	6.62e-06	1.57 (95% CI: 1.29-1.91)
ES	I	Hypertrophic cardiomyopathy	11.82 (11.10-12.53)	420/439981	High	4.61e-02	1.77e-01	1.15 (95% CI: 0.94-1.40)
ES	IL	Hypertrophic cardiomyopathy	11.82 (11.10-12.53)	420/439981	High	6.69e-07	5.07e-08	1.72 (95% CI: 1.41-2.08)
ES	AL	Hypertrophic cardiomyopathy	11.82 (11.10-12.53)	420/439981	High	6.26e-07	5.35e-06	1.57 (95% CI: 1.29-1.91)
ES	A	Hypertrophic cardiomyopathy	11.82 (11.10-12.53)	420/439981	High	1.32e-03	6.22e-03	1.32 (95% CI: 1.08-1.61)
ES	AS	Hypertrophic cardiomyopathy	11.82 (11.10-12.53)	420/439981	High	2.91e-06	4.67e-05	1.50 (95% CI: 1.24-1.83)
ED	IS	Hypertrophic cardiomyopathy	11.82 (11.10-12.53)	420/439981	High	8.40e-01	5.76e-01	1.06 (95% CI: 0.86-1.30)
ED	I	Hypertrophic cardiomyopathy	11.82 (11.10-12.53)	420/439981	High	1.32e-03	3.66e-03	1.34 (95% CI: 1.10-1.62)
ED	IL	Hypertrophic cardiomyopathy	11.82 (11.10-12.53)	420/439981	High	3.56e-01	1.24e-01	1.17 (95% CI: 0.96-1.43)
ED	AL	Hypertrophic cardiomyopathy	11.82 (11.10-12.53)	420/439981	High	3.56e-01	1.07e-01	1.18 (95% CI: 0.96-1.45)
ED	A	Hypertrophic cardiomyopathy	11.82 (11.10-12.53)	420/439981	High	1.35e-03	4.24e-04	1.42 (95% CI: 1.17-1.73)
ED	AS	Hypertrophic cardiomyopathy	11.82 (11.10-12.53)	420/439981	High	2.15e-01	1.64e-01	1.15 (95% CI: 0.94-1.41)

7. Line 262: “remaining 367,831 individuals without CMR imaging data”. The description of the source of individuals without CMR imaging is unclear. Please provide a detailed explanation of the screening process for this population.

Response: Thank you for the comment. We have revised it in the revised manuscript.

In line 433-440, we now state:

Among the initial 456,937 individuals without CMR data, we excluded individuals without genotype data, resulting in a final sample size of 442,889 individuals. After accounting for missing information on smoking and alcohol consumption, the sample size was further reduced to 440,085 individuals. Additionally, in subsequent analyses focusing on specific diseases, we further excluded individuals who had the corresponding disease prior to their enrollment in the UKB. For instance, in the case of HCM, we excluded 104

individuals who had HCM before their enrollment, resulting in a final sample size of 439,981 individuals.

Minor issues

1. Line 243: “rage”, maybe replace by “range”.

Response: Thank you for the comment. We have revised it in the revised manuscript.

In line 260, we now state:

beta range: 0.45 to 3.10, $P < 0.01$.

2. The author only presented the r^2 of phenotype correlations in Supplementary Fig 2 without significant annotations. Please add annotations of P-values of phenotype correlations.

Response: Thank you for the comment. We have revised it in the revised Supplementary Figures.

3. The authors performed Mendelian randomization analyses using two-sample methods to investigate the causal association between LVRWT traits and cardiovascular diseases. The GWAS summary information of MR analyses should be Provided.

Response: Thank you for the comment. We have revised it in the revised Supplementary Tables.

4. While the manuscript was understandable, the English writing could be improved with some refinement.

Response: Thank you for the comment. We have revised it in the revised manuscript.

Reviewer #3:

Ning et al described the genetic architecture of left ventricular regional wall thickness (LVRWT) traits. The authors identified 72 genetic loci associated with LVRWT traits. They indicated that risk variants may regulate the expression of genes by activating chromatin states, which enhanced the understanding of the mechanisms underlying GWAS risk loci. Interestingly, the authors build polygenic risk scores (PRS) weighting the genetic dosage from each LVRWT trait GWAS, which was found to have significant relationships with incident hypertrophic cardiomyopathy (HCM). The paper is well-performed, and I have only a few comments regarding its acceptance.

Response: Thanks for the positive comments. We have addressed the concerns in the following responses.

Major issues

1. The authors mentioned in lines 401-402 that individuals with a history of previous myocardial infarction or a diagnosis of heart failure were excluded. Please explain the exclusion.

Response: Thank you for your question. The exclusion of individuals with a history of previous myocardial infarction or a diagnosis of heart failure was done to ensure the homogeneity of our study population and minimize potential confounding factors. This approach is consistent with strategies used in other studies to enhance the robustness and interpretability of findings (Ahlberg et al. *Eur Heart J.* 2021; 42(44):4523-4534; Aung et al. *Circulation.* 2019; 140(16):1318-1330).

2. Protein modification includes various forms such as methylation, acetylation, phosphorylation, etc. They can affect the transcription, replication, and repair of genes by changing the structure and function of chromatin. It is an important regulatory mechanism in biological processes such as cell differentiation, development, and disease occurrence. In this article, the authors chose H3K4me1, H3K4me3, H3K27ac, H3K36me3 and H3K9me3 to examine the risk variants enrichment situation. There are hundreds of histone modifications available. Please explain in detail why the authors chose these histone modifications.

Response: Thank you for your question. These histone modifications were selected based on their established roles in regulating chromatin structure and gene expression. H3K4me1 and H3K4me3 are associated with enhancers and promoters, respectively, and are involved in transcriptional activation. H3K27ac is a mark of active enhancers and promoters, reflecting active transcription and regulatory elements. H3K36me3 is linked to transcription elongation and gene expression, while H3K9me3 is often associated with gene silencing and heterochromatin formation. We opted for these five histone

modifications due to their well-defined roles in gene regulation and their relevance to cardiovascular traits and diseases.

3. In Figure 3, the authors provided rich annotations for the obtained risk variants, including location annotations, histone enrichment analysis, and disease enrichment analysis. These annotations are meaningful, especially as Figure 3D shows that risk variants are significantly enriched in some heart-related diseases. Currently, single-cell-related research is booming, and the authors may consider examining whether these risk variants are specifically enriched in certain heart or myocardial-related cells, which may further explain the findings in Figure 3D.

Response: Thank you for your insightful suggestion. You have emphasized the significance of single-cell-related research in understanding complex genetic associations. However, it's important to note that current single-cell technologies mainly encompass sc-RNA-seq and sc-ATAC-seq, and the sequencing of single-cell genotypes remains a technical challenge. As a result, at present, we lack the necessary data to investigate whether these risk variants are enriched in specific cardiac cell types using single-cell genomics. As the field of single-cell genomics evolves and new techniques emerge, it holds great potential to provide deeper insights into the specific cell types and mechanisms underlying the observed genetic associations with heart-related diseases. Your suggestion will certainly guide our future research directions as we strive to unravel the finer details of genetic contributions to cardiac remodeling and related conditions. We have declared the limitation in the Discussion section of the revised manuscript.

In line 405-409, we now state:

Lastly, although our findings provide valuable insights into the genetic basis of LVRWT, the complexities of genetic regulation and functional mechanisms within specific cardiac cell types might require more advanced methodologies, such as emerging single-cell genomics techniques, to fully unravel the intricate genetic networks underlying LVRWT and related cardiac remodeling.

4. The authors investigated the genetic correlation among 12 LVRWT traits in Figure 5A and found that most of the LVRWT traits are strongly correlated. However, in Figure 5B, the authors further explored the genetic correlation between these LVRWT traits and 10 traits related to cardiac structure and function. Why did the author take this step and not skip directly to the procedure in Figure 5C?

Response: Thank you for your question. This approach allowed us to elucidate the intricate relationships between LVRWT and other cardiac structural and functional traits, contributing to a comprehensive understanding of the genetic underpinnings of LVRWT

traits. The strong positive genetic correlations observed between LVRWT traits and 10 traits highlight the shared genetic influences contributing to these cardiac phenotypes.

5. Cardiomyopathy was included as one of the 11 cardiovascular diseases analyzed in the genetic correlation study of left ventricular wall thickness measures, as described in lines 482-484. However, the authors specifically chose to investigate dilated cardiomyopathy and hypertrophic cardiomyopathy as distinct subtypes. Please provide a detailed explanation for why selected these two cardiomyopathy subtypes for analysis, despite inclusion under the broader category of cardiomyopathy?

Response: Thank you for your question. Dilated cardiomyopathy (DCM) and hypertrophic cardiomyopathy (HCM) are among the most prevalent and clinically significant subtypes within the cardiomyopathy spectrum. Despite their shared category, they display notable genetic heterogeneity. Focusing on them individually aims to uncover subtype-specific genetic associations and better comprehend underlying mechanisms.

6. The authors constructed PRS for 12 LVRWT traits and predicted the incidence of HCM in non-CMR populations, and found that some of the PRS for LVRWT were predictive of the occurrence of HCM. However, it is unclear whether these PRS predict other heart diseases, such as dilated cardiomyopathy.

Response: Thank you for raising this important question. Following your recommendation, we conducted additional analyses to assess whether the constructed PRS for LVRWT traits also have predictive capabilities for other heart diseases, such as dilated cardiomyopathy. The outcomes of these analyses have been included in Supplementary Table 19. These supplementary findings contribute to a more comprehensive understanding of the potential broader implications of the PRS beyond hypertrophic cardiomyopathy.

In line 290-292, we now state:

We also assessed whether the constructed PRS for LVRWT traits also have predictive capabilities for other 10 CVDs (**Supplementary Table 19**).

Minor issues

1. The abbreviation 'EA' in the title row of Table 1 spans two lines, please make corrections.

Response: Thank you for the comment. We have revised it in the revised Table 1.

2. It is better to label the p-values in Figure 7CD as "log-rank P".

Response: Thank you for the comment. We have revised it in the revised Figure 7.

3.Minor Allele Frequency (MAF) can be included as an additional column in Supplementary Table 18.

Response: Thank you for the comment. We have revised it in the revised Supplementary Table 18.

4.The manuscript comprises a vast array of diverse information. The author should enhance the coherence of the text.

Response: Thank you for the comment. We have revised it in the revised manuscript.

Reviewer #4:

The authors performed a GWAS on left ventricular regional wall thickness LVRWT, followed by a substantial number of post-GWAS analyses. By and large while the methods are fairly robust (and standard, which is a good thing), the manuscript lacks important details on outcome definitions and data sources.

Response: We greatly appreciate your diligent review of our article. Your expert insights have been immensely valuable in refining our work. We have thoroughly addressed the points you raised, and the specific responses are provided below.

Major comments:

1. The authors presumably use aggregated GWAS data for their the genetic correlation and the Mendelian randomization analyses. The source of these data is however not provided at all in their manuscript, or at least not in the data availability section, the methods or the figure legends. Without this information the reader is left wondering how these estimate magically appear, and above all cannot attempt to replicate the presented findings.

Response: Thank you for bringing up this important point. We acknowledge the significance of providing clear information about the source of the GWAS data used in our genetic correlation and Mendelian randomization analyses. To address this concern, we have included details about the GWAS data source in **Supplementary Table 3**.

2. Similarly, but distinct, for the PRS analysis the authors attempt to predict HCM onset in UKB participants without CMR data. Great idea of course. However, the authors do not explain how HCM was defined, whether this is a HCM diagnosis (I hope) or merely carriership? Do the HCM cases pertain to incidence (after UKB enrolment), or also include prevalent cases? Given that PRS is about prediction, including prevalent cases is of course irrelevant. The major problem of course is that I do not have any idea if this comment on prevalent or incidence is important here, the authors have not provided any explanation. At the minimal please include a code list how HCM was defined and at what time relevant to UKB enrolment.

Response: Thank you for your question. HCM is defined based on the ICD10 codes I42.1 and I42.2, as indicated by UKB Field ID 41270. These codes correspond to the disease phenotype of HCM as indicated in Supplementary Table 20. We would like to emphasize that our approach of focusing on incident cases of HCM, which occurred after participants' enrollment in the UKB (according to UKB Field ID 41280), aligns with approaches employed in established studies (Pirruccello et al. *Nat Genet.* 2022;54(1):40-51; Pirruccello et al. *Nat Genet.* 2022;54(6):792-803). This approach ensures that our PRS analysis focuses on predicting the onset of HCM, providing meaningful insights into

the predictive potential of our PRS models. We have revised it in the revised manuscript.

In line 428-429, we now state:

Disease information based on the UKB Field ID and date of first in-patient diagnosis is provided in **Supplementary Table 20**.

Supplementary Table 20. Disease definitions

Disease	Data fields	Field names	Data codes	Field of date of first in-patient diagnosis
Hypertrophic cardiomyopathy	41270	Diagnoses - ICD10	I42.1 & I42.2	41280
Hypertension	131287	Source of report of I10 (essential (primary) hypertension)	I10	131286
Pulmonary hypertension	41270	Diagnoses - ICD10	I27.0 & I27.2	41280
Dilated cardiomyopathies	41270	Diagnoses - ICD10	I42.0	41280
Cardiomyopathy	131339	Source of report of I42 (cardiomyopathy)	I42	131338
Heart failure	131355	Source of report of I50 (heart failure)	I50	131354
Atrial fibrillation	131351	Source of report of I48 (atrial fibrillation and flutter)	I48	131350
Angina pectoris	131297	Source of report of I20 (angina pectoris)	I20	131296
Myocardial infarction	42001	Source of myocardial infarction report	-	42000
Chronic ischaemic heart disease	131307	Source of report of I25 (chronic ischaemic heart disease)	I25	131306
Ischemic stroke	42009	Source of ischaemic stroke report	-	42008

3. On the presentation of the PRS, while I understand that the presented scores significantly associate with HCM (however defined), and the author do provide HR for low, intermediate and high PRS groups, this presentation has been shown to be insufficient in the statistical literature for decades; see Royston BMC MRM 2013. Please, at least additional (if not substitute) provide measures of discrimination such as the c-statistics or AUC. Based on the presented results I am confident the c-statistic/AUC will be extremely close to 0.50 suggesting the benefit of these PRS are likely minimal. In all fairness, most genetic PRS studies to present their results without c-stat/AUC, presumably because these are simply fairly small and unlike the HR not bounded between 0 and 1.

Response: Thank you for your question. We acknowledge your perspective that in many PRS studies, the C-statistic/AUC measures are often omitted due to their relatively modest values, which may limit the discriminatory power. For instance, in the studies that constructed PRS for cardiac structural phenotypes based on CMR data (Pirruccello et al. *Nat Genet.* 2022;54(1):40-51; Pirruccello et al. *Nat Genet.* 2022;54(6):792-803), the C-statistic/AUC values were not calculated. However, we agree that providing additional measures can enhance the interpretation of PRS performance.

In response to your recommendation, we calculated the C-statistic for our PRS about HCM. The obtained C-statistic reached 0.57, suggesting that our PRS does possess some discriminatory power in distinguishing individuals with varying levels of HCM risk. It

is worth noting that the addition of ES-IS PRS to the clinical risk factors yielded a modest yet significant improvement in C-statistic (1%, **Figure R1**). This finding aligns with the observation that the high PRS group shows a higher risk for HCM, further reinforcing the effectiveness of our PRS in identifying individuals at elevated risk.

While we acknowledge that achieving higher C-statistic values can be challenging in genetic PRS studies, particularly in complex phenotypes like HCM, we believe that this measure provides valuable information about the potential clinical utility of our PRS.

Figure R1. C-statistic results of 12 LVRWT PRS for HCM prediction.

C-statistics are based on 11.8-year follow-up events from Cox regression models of listed variables. Clinical risk factors includes age , sex, BMI, smoking status, alcohol intake frequency and PC1-10 variables in its risk estimation.

4. Have the authors considered combining multiple trait specific PRS into a stacked model to predict HCM. See <https://www.medrxiv.org/content/10.1101/2022.09.01.22279477v1>

Response: Thank you for mentioning the study by Dziopa et al. The authors developed predictive models for various CVD outcomes using a multivariable "stacked" polygenic score (PGS) approach, with the aim of improving the discriminatory power of these models.

In our study, we constructed PRS for each of the 12 LVRWT traits. This choice was based on the distinct relevance of each LVRWT trait to specific cardiac conditions. For instance, the inferoseptal LVRWT might exhibit a specific association with conditions like HCM. Notably, inferoseptal hypertrophy with a thickness greater than 15 mm was a major

component of the diagnostic criteria for HCM (Gersh et al. *Circulation*. 2011;124(24):2761-96). By analyzing each LVRWT trait separately, we aimed to assess the unique impact of each LVRWT trait on the occurrence of HCM, potentially shedding light on the specific cardiac regions contributing to the development of the condition. While we appreciate the suggestion of a stacked model for PRS, our focus on individual LVRWT traits helps us capture their distinct associations with HCM, leading to a more detailed and specific analysis of the genetic factors influencing different aspects of cardiac remodeling in relation to hypertrophic cardiomyopathy.

5. To what extent are the LVRWT measurements correlated? Could the authors please include a principle component analysis and identify how many PCs were necessary to explain 90% of the variability in the LVRWT measurements. This would give an indication of the multiple testing threshold.

Response: We appreciate your question. We conducted a PCA on the LVRWT measurements and the analysis revealed that utilizing five PCs accounted for over 90% of the variability in the LVRWT measurements. This information provides insights into the dimensionality of the data and helps establish a reasonable multiple testing threshold. We have revised it in the revised manuscript.

In line 135-137, we now state:

Furthermore, to gain insight into the dimensionality of the data, we conducted a principal component analysis (PCA) and identified that utilizing five principal components explained over 90% of the variability in the LVRWT measurements (**Supplementary Fig. 2B**).

B

Supplementary Fig. 2. Phenotypic correlation and PCA analysis of 12 LVRWT traits.

B. the results of Principal Component Analysis (PCA) applied to the 12 LVRWT traits. Through PCA, we projected the original traits onto 5 principal components, which collectively explain 91.6% of the total variance. Abbreviation: LVRWT, LV regional wall thickness; MSMM, Myocardial Segmentation and Measurement Method; IS, inferoseptal;

I, inferior; IL, inferolateral; AL, anterolateral; A, anterior; AS, anteropetal; ES, end systole; ED, end diastole.

6. Could the authors please identify which signals passed the canonical GWAS significance threshold corrected for the number of PCs (see above).

Response: Thank you for your suggestion. We have taken your advice into consideration and recalculated the GWAS significance threshold by adjusting it with the number of PCs. We have included the results in **Table 1**, indicating variants with $P < 1.0 \times 10^{-8}$ using an asterisk (*). Additionally, in **Figure 2**, we've marked the threshold of $P < 1.0 \times 10^{-8}$ with a red line. We have revised it in the revised manuscript.

In line 154-155, we now state:

Furthermore, following the five PCs correction for multiple tests, we observed that 47 variants remained significant ($P < 1.0 \times 10^{-8}$, Table 1).

Figure 2

Fig. 2. Manhattan plots of genome-wide association studies results for 12 LVRWT phenotypes.

Manhattan plots show the chromosomal position on the x-axis and the $-\log_{10}(P)$ on the y-axis for each LVRWT phenotype. The black dashed line indicates the genome-wide significance threshold at $P < 5 \times 10^{-8}$, while the red dashed line represents the significance level after multiple corrections ($P < 1.0 \times 10^{-8}$). Loci that contain variants with $P < 1 \times 10^{-8}$ were labeled with the name of the nearest gene. P values are two sided based on the chi-squared test statistics in BOLT-LMM software. Abbreviation: LVRWT, LV regional wall thickness; IS, inferoseptal; I, inferior; IL, inferolateral; AL, anterolateral; A, anterior; AS, anterospetal

7. Regarding the MR analyses and the multiple testing correction it seems the authors adjusted the p-value threshold for the number of methods used (5). Given that these methods all use the same (or subset of the same) data, and are therefore correlated, such a correction is inappropriate. Instead please correct for the number of outcomes (11) times the number of PCs necessary to explain 90% of the variability in the CMR traits.

Response: Thank you for your valuable suggestion. Following your suggestion, we have recalculated the p-value threshold. We have updated Figure 6, where the red coloring now indicates variants with a significance threshold of $P < 0.05/55$, while the green coloring represents variants with a significance threshold of $P < 0.05$. We have revised it in the revised manuscript.

Fig. 6. Causal effects between LVRWT phenotypes and CVDs using Mendelian Randomization.

The GWAS data sources used for Mendelian Randomization are detailed in Supplementary Table 3. Effect sizes are presented as ORs per standard deviation increment in LVRWT in relation to the risk of CVDs. The horizontal bars represent 95% confidence intervals (95% CIs), with color-coding indicating significance levels: Red denotes multiple correction for significance ($P < 0.05/55$), Green signifies nominal significance ($P < 0.05$), and faded grey indicates non-significant correlations. For two sample MR analyses, we used MR-Egger regression, weighted median and mode-based estimations as sensitivity analyses, along with the inverse-variance weighted technique as our major model. Abbreviation: LVRWT, LV regional wall thickness; CVDs, cardiovascular diseases; IS, inferoseptal; I, inferior; IL, inferolateral; AL, anterolateral; A, anterior; AS, anterospetal; OR, Odds ratio; SD, standard deviation.

8. The authors do not seem to be planning on sharing their aggregated GWAS results (based on the data availability section). This would be extremely poor conduct, especially because most of their analyses just discussed used other peoples shared GWAS results. The authors should immediately and fully share all their aggregate results, which do not have any privacy concerns. Please do not share them via some weird online shiny app or otherwise “data friendly” website – which is a form of data obfuscation, and simply share the gzipped files, include the effect allele, beta coefficients, standard error and p-values. Here is the GWAS catalog deposition link: <https://www.ebi.ac.uk/gwas/deposition>

Response: Thank you for the comment. In line with your recommendation, we have promptly deposited our self-generated GWAS results, including effect alleles, beta coefficients, standard errors, and p-values, into the GWAS catalog deposition (<https://www.ebi.ac.uk/gwas/deposition>) under GCP ID: GCP000711. Since our data falls under the category of unpublished data submissions, each summary statistics file will be made available on ftp server within 48 hours. Due to this recent deposit, it is possible that the GWAS results are not yet accessible in the catalog. Additionally, we have included the publicly available GWAS results in Supplementary Table 3 as part of our commitment to transparency and data sharing. We have revised it in the revised manuscript.

In line 759-762, we now state:

Data sources of publicly available GWAS results were listed in **Supplementary Table 3**. The GWAS summary statistics generated in this study have been deposited in the Human Genome Research Institute GWAS Catalog under GCP ID: GCP000711.

9. The author seem to have overlooked a few recent GWAS on CMR trait which are relevant here:

<https://pubmed.ncbi.nlm.nih.gov/36598836/>

<https://www.science.org/doi/10.1126/sciadv.add4984>

Response: Thank you for the comment. We have compared our GWAS results with those from Aung’s study, as you mentioned. In their study, they identified a total of 21 loci related to CMR traits. Notably, our analysis has demonstrated a remarkable replication of their findings, with 11 out of those 21 loci being consistently identified in our research, showcasing consistency in findings across different investigations. Importantly, our investigation has further unveiled 25 additional loci that were not part of Aung’s study. These new loci include significant candidate genes such as *NMB* (Shah et al. *Circ Cardiovasc Genet.* 2011;4(6):626-35) and *WNT3* (Ren et al. *Front Cardiovasc Med.* 2021; 8:675222), which have established connections with inherited cardiovascular diseases. The comparison of our findings and Aung’s study is summarized in Table R1.

Schmidt's study you referred to is particularly interesting as it delves into the influence of proteins on CMR imaging metrics. While many traits in their study were based on volume-related indicators, our research has focused on identifying 12 distinct LVRWT traits. In the future, we envision applying a similar approach to explore how proteins impact wall thickness, potentially shedding light on their role in relevant cardiac conditions.

Table R1. Comparison of loci between the current study and Aung et al.

Loc	Compare	Annotated genes in our paper	Aung gene
17q21.31	common	LRRC37A4P, ARL17B, DND1P1, KANSL1-AS1, LRRC37A2, MAPT, ARL17A, ARHGAP27, PLEKHM1, KANSL1, LRRC37A, NSF, MAPT-AS1, CRHR1, SPPL2C, STH	MAP3K14
15q25.3	common	ZNF592, NMB, ALPK3, CSPG4P12, SEC11A	ALPK3
12q21.33	own	ATP2B1, POC1B, GALNT4, POC1B-GALNT4	-
17q21.32	own	WNT3, GOSR2, RPRML, WNT9B, CDC27, MYLA, ITGB3, SNX11, MRPL10, RP11-156P1.2	-
1p36.22	own	CLCN6, AGTRAP, NPPA, NPPB, MTHFR	-
10q22.2	common	SYNPO2L, ECD, DNAJC9, MRPS16, TTC18, MYOZ1, AGAP5, SEC24C, FUT11, PLAU, C10orf55, ZSWIM8, RP11-574K11.31, NDST2, CAMK2G, DUSP8P5	SYNPO2L
10q26.11	own	BAG3	-
11p11.2	common	SPI1, ACP2, MADD, SLC39A13, RAPSN, NDUFS3, C1QTNF4, MTCH2, PTPRJ, MYBPC3, PSMC3, CELF1, AGBL2, FNBP4, NUP160	RAPSN
15q22.31	own	PARP16	-
15q26.3	own	IGF1R	-
17q23.2	own	BCAS3	-
18q11.2	own	CTAGE1	-
18q12.1	common	MAPRE2	MAPRE2
1p13.1	own	CASQ2	-
1p36.32	own	PRDM16	-
1q22	own	ADAM15, DCST2, EFNA1, EFNA3, MUC1, RIT1, THBS3	-
1q42.2	own	CAPN9, AGT, C1orf198	-
20p12.2	own	SLX4IP	-
21q22.3	own	LINC00322	-
22q11.23	common	SMARCB1, C22orf43, VPREB3, MMP11, DERL3	SMARCB1
22q12.1	common	MYO18B	MYO18B
22q12.2	own	SMTN	-
2p14	own	CEP68, ACTR2, RAB1A	-
3p14.1	own	MITF, FRMD4B	-

4q21.21	own	FGF5	-
4q34.2	own	-	-
5q31.1	own	HSPA4	-
5q35.1	own	RPL7AP33, BNIP1, NKX2-5	-
6p12.1	own	HCRTR2, FAM83B	-
6p21.2	common	CDKN1A	CDKN1A
8p21.3	common	XPO7, DMTN, BIN3	XPO7
8p23.1	own	RP11-10A14.4, ERI1, CLDN23, MFHAS1, MSRA, GATA4, FDFT1	-
8q23.1	own	HMGB1P46	-
8q24.13	common	MTSS1, SQLE, LINC00964	LINC00964
8q24.3	common	PTK2, AGO2	PTK2
9q32	own	ATP6V1G1	-
1q32.3	Aung	-	PROX1
2p16.1	Aung	-	EFEMP1
3p13	Aung	-	MITF
3q26.31	Aung	-	FNDC3B
6p21.1	Aung	-	TFEB
12p12.2	Aung	-	LINC02398
12q24.23	Aung	-	PXN
14q32.13	Aung	-	GSC
17q22	Aung	-	HIF
18q12.2	Aung	-	FHOD3

Reviewer #5:

1. It is a big leap of faith in saying that a model trained on ACDC dataset is good enough to segment the UKB population data. Especially when the authors mention the use of pathological and healthy images for ACDC and UKB is largely a healthy population. And that is why it is hard to verify the accuracy of the proposed DLA model. Authors should put effort in addressing this issue and help in bridging this gap. Some notes:

a. The details of the deep learning architecture are quite poorly explained by the authors even in the supplementary notes. It would be useful to include the more details like number of hidden layers, some example segmentations that show accuracy of the models.

Response: Thanks for your comments. The number of hidden layers of DLANet is 166, and the number of parameters is 20,576,932. Example segmentation results on UKB dataset are shown in Supplementary Fig 14. We can see that the segmentation results from DLANet have strong consistency with the ground-truth labels, which demonstrate the effectiveness of the DLANet. We have added more details in the methods section of revised manuscript and Supplementary Methods.

Supplementary Fig 14. Illustration of segmentation results of DLANet on the UKB dataset.

First row: the original short axis images. Second row: manual labels by the doctors. Last row: the prediction results generated by the DLANet. Green: LV myocardium, blue: LV cavity

b. The testing dataset is quite small for the DLA model, I wonder how sensitive the LV wall thickness calculations are to the number of images as input for DLA architecture training.

Response: Thanks for your comments. There are 1902 annotated images in the ACDC dataset. Among them, 1420 images are used for training, 100 images for validating and

the left 382 short axis images for testing. Within the 382 testing images, 80 images belong to the mid-cavity slices. Additionally, we included 500 mid-cavity images from the UKB dataset for independent evaluation. To observe the sensitivity of the MAE of myocardial thickness, we tested the DLANet on both the ACDC and UKB datasets with different numbers of training images, i.e. 25%, 50%, 75%, and 100% of the training data. As shown in Supplementary Table 24 and Supplementary Table 25, even when training the DLANet with only 25% of the images, the MAEs of these two datasets are only at 0.945 mm and 0.807mm. While there are some improvements as the amount of training data increased, the differences between models trained with 25% and 100% of the data are not significant. These findings have two implications. On one hand, they demonstrate that the DLANet has the potential to generate accurate results even with a small number of input images. On the other hand, the mid-cavity images of the heart have high quality and strong contrast between the LV's endocardium and epicardium, which makes segmentation and quantification easier compared to slices located elsewhere. We have added the details in the methods section of revised manuscript and Supplementary Methods.

Supplementary Table 24. Testing results of DLANet on ACDC dataset by using different number of training data.

Percent of training-data	Dice (\uparrow)		HD (mm, \downarrow)		MAE (mm, \downarrow)
	LVC	LVM	LVC	LVM	
25%	0.963 \pm 0.027	0.905 \pm 0.032	2.730 \pm 1.244	3.319 \pm 1.210	0.945 \pm 0.367
50%	0.967 \pm 0.020	0.912 \pm 0.026	2.726 \pm 1.112	3.284 \pm 1.122	0.898 \pm 0.323
75%	0.966 \pm 0.026	0.913 \pm 0.025	2.644 \pm 0.827	3.281 \pm 0.997	0.899 \pm 0.296
100%	0.968\pm0.018	0.915\pm0.025	2.545\pm0.787	3.171\pm0.935	0.853\pm0.304

The total number of training data (100%) is 1420. The best results are highlighted in bold. Aberrations: Dice: Dice coefficient, HD: Hausdorff Distance, MAE: mean absolute errors. LVC, left ventricular cavity; LVM, left ventricular myocardium

Supplementary Table 25. Testing results of DLANet on UKB dataset by using different number of training data.

Percent of training-data	Dice (\uparrow)		HD (mm, \downarrow)		MAE (mm, \downarrow)
	LVC	LVM	LVC	LVM	
25%	0.969 \pm 0.023	0.937 \pm 0.027	2.195 \pm 1.115	2.613 \pm 1.303	0.807 \pm 0.411
50%	0.971 \pm 0.025	0.946 \pm 0.021	1.940 \pm 1.146	2.340 \pm 1.063	0.761 \pm 0.371

75%	0.971±0.024	0.947±0.021	2.015±1.068	2.390±1.098	0.735±0.387
100%	0.974±0.021	0.949±0.020	1.865±0.932	2.236±1.014	0.663±0.343

Models are trained on ACDC dataset. The best results are highlighted in bold. Aberrations: Dice: Dice coefficient, HD: Hausdorff Distance, MAE: mean absolute errors. LVC, left ventricular cavity; LVM, left ventricular myocardium

c. Please include the time-performance of the DLA method on the said GPU you have used for segmentation.

Response: Thanks for your comments. One frame of a patient with all slices in the size of (8-12)*256*256 will take about 0.5s on one NVIDIA GeForce GTX 1080 Ti GPU at the inference stage. We have also added the description in the methods section of revised manuscript and Supplementary Methods.

d. The following sentence seems a bit unscientific: “As we expected, after a visual evaluation of segmentation results on random 100 images by an experienced doctor, it could be considered that the model also performed well on the UKB dataset.”

Response: Thanks for your comments. To demonstrate the DLANet have ability to perform well also on the UKB dataset, we resorted to a doctor who has 7 years of clinical experience to label the LV cavity and LV myocardium. A total of 50 patients were annotated, with 5 randomly selected frames for each patient, and in 8-14 slices per frame. Among these annotations, we focused on selecting the mid-cavity slices to form the UKB testing set, and result in total of 500 short-axis images. The testing results are reported in Supplementary Table 23, where four different methods are compared. From these results, it is evident that the DLANet achieved good performance with an average Dice coefficient of 0.962, an average Hausdorff distance of 2.051 mm, and a MAE of 0.663. These indicate that the DLANet already meets the requirements of practical applications. We have added the details in the methods section of revised manuscript and Supplementary Methods.

i. I think a more scientific approach is required to quantify the performance. For example, the clinicians can blindly segment the images and then you could use some sort of score to compare the segmentations. I think the authors are assuming that clinical segmentations are ground truth, which is understandable so a direct comparison of segmentations of untrained images is warranted.

Response: Thanks for your comments and please refer to the response for the question d.

References

1. Aung N, *et al.* Genome-Wide Analysis of Left Ventricular Image-Derived Phenotypes Identifies Fourteen Loci Associated With Cardiac Morphogenesis and Heart Failure Development. *Circulation* **140**, 1318-1330 (2019).
2. Perez-Pelegri M, Monmeneu JV, Lopez-Lereu MP, Maceira AM, Bodi V, Moratal D. End-systole and end-diastole detection in short axis cine MRI using a fully convolutional neural network with dilated convolutions. *Comput Med Imaging Graph* **99**, 102085 (2022).
3. Roth GA, *et al.* Global Burden of Cardiovascular Diseases and Risk Factors, 1990-2019: Update From the GBD 2019 Study. *J Am Coll Cardiol* **76**, 2982-3021 (2020).
4. Ahlberg G, *et al.* Genome-wide association study identifies 18 novel loci associated with left atrial volume and function. *Eur Heart J* **42**, 4523-4534 (2021).
5. Aung N, *et al.* Genome-wide association analysis reveals insights into the genetic architecture of right ventricular structure and function. *Nat Genet* **54**, 783-791 (2022).
6. Gersh BJ, *et al.* 2011 ACCF/AHA guideline for the diagnosis and treatment of hypertrophic cardiomyopathy: executive summary: a report of the American College of Cardiology Foundation/American Heart Association Task Force on Practice Guidelines. *Circulation* **124**, 2761-2796 (2011).
7. Pirruccello JP, *et al.* Deep learning enables genetic analysis of the human thoracic aorta. *Nat Genet* **54**, 40-51 (2022).
8. Pirruccello JP, *et al.* Genetic analysis of right heart structure and function in 40,000 people. *Nat Genet* **54**, 792-803 (2022).
9. Shah S, *et al.* Four genetic loci influencing electrocardiographic indices of left ventricular hypertrophy. *Circ Cardiovasc Genet* **4**, 626-635 (2011).
10. Ren H, *et al.* WNT3A rs752107(C > T) Polymorphism Is Associated With an Increased Risk of Essential Hypertension and Related Cardiovascular Diseases. *Front Cardiovasc Med* **8**, 675222 (2021).

REVIEWERS' COMMENTS

Reviewer #1 (Remarks to the Author):

All questions have been addressed.

Reviewer #2 (Remarks to the Author):

Authors have properly responded my concerns. I have no additional comments.

Reviewer #3 (Remarks to the Author):

The authors have addressed all my comments. The quality of the manuscript has significantly improved in my opinion. I feel that the results are novel, interesting and important.

Reviewer #4 (Remarks to the Author):

Please include the additional figure R1 “C-statistic results of 12 LVRWT PRS for HCM prediction”, and discuss its implication in the manuscript. By leaving this out of the actual paper and refusing to alter their interpretation based on the metrics the authors are seemingly fobbing me off. I will give a brief overview of where the authors incorrectly interpret the relevance of their PRS.

The authors now include c-statistic to directly measure discriminative performance, showing a very low discriminative performance close to 0.50 (flipping a coin), and sometimes none at all. Of note the models with a c-statistic of 0.60 include clinical variables, and the model combining the genetic and clinical data is not statistically different than the clinical only model. So really they show the PRS does NOT add anything. However, their conclusions still state:

“The polygenic risk score of inferoseptal LVRWT at end systole had a significant relationship with incident HCM, with favorable discrimination performance for high-risk HCM populations. “

“providing valuable risk stratification guidance for HCM surveillance.”

Less arrogantly, but still wrong

“Furthermore, we show that PRSs derived from LVRWT traits predict incident HCM in individuals without CMR data”

Similarly,

“ . We showed that PRSs derived from LVRWT traits predict incident HCM. This suggested that the genetic basis of LVRWT traits may provide valuable risk stratification guidance to identify high-risk individuals for early screening of HCM”

With a c-statistic close to 0.50 I would suggest to merely say the PRSs associated with HCM, but certainly did not provide any bases for accurate prediction. You could of course argue that the PRS very very very poorly predicted HCM carriership – but I expect the authors would rather not say that, hence my suggestion for association instead.

Bottom line please remove all these hyperbolic and factually incorrect interpretations saying that your PRSs were discriminatory in any sense aside from the mere fact that some of them marginally excluded a null-value of 0.05. Truthfully you found that they provide very weak information on HCM carriership or not, and any model using the PRS would need to include substantially more information to provide any discriminative ability of note (say 0.70, although this might also be low for actual classification).

When I suggest remove, please check the entire document and not just the few sentences I cited.

Reviewer #5 (Remarks to the Author):

I am satisfied with the authors' response.

Reviewer #1 (Remarks to the Author):

All questions have been addressed.

Response: We appreciate your feedback and are glad to hear that your concerns have been properly addressed.

Reviewer #2 (Remarks to the Author):

Authors have properly responded my concerns. I have no additional comments.

Response: Thank you for your understanding, and we're pleased that your concerns have been resolved.

Reviewer #3 (Remarks to the Author):

The authors have addressed all my comments. The quality of the manuscript has significantly improved in my opinion. I feel that the results are novel, interesting and important.

Response: Thank you for your feedback and understanding. We appreciate your satisfaction with our response.

Reviewer #4 (Remarks to the Author):

Please include the additional figure R1 "C-statistic results of 12 LVRWT PRS for HCM prediction", and discuss its implication in the manuscript. By leaving this out of the actual paper and refusing to alter their interpretation based on the metrics the authors are seemingly fobbing me off. I will give a brief overview of where the authors incorrectly interpret the relevance of their PRS.

The authors now include c-statistic to directly measure discriminative performance, showing a very low discriminative performance close to 0.50 (flipping a coin), and sometimes none at all. Of note the models with a c-statistic of 0.60 include clinical variables, and the model combining the genetic and clinical data is not statistically different than the clinical only model. So really they show the PRS does NOT add anything. However, their conclusions still state:

"The polygenic risk score of inferoseptal LVRWT at end systole had a significant relationship with incident HCM, with favorable discrimination performance for high-risk HCM populations. "

“providing valuable risk stratification guidance for HCM surveillance.”

Less arrogantly, but still wrong

“Furthermore, we show that PRSs derived from LVRWT traits predict incident HCM in individuals without CMR data”

Similarly,

“We showed that PRSs derived from LVRWT traits predict incident HCM. This suggested that the genetic basis of LVRWT traits may provide valuable risk stratification guidance to identify high-risk individuals for early screening of HCM”

With a c-statistic close to 0.50 I would suggest to merely say the PRSs associated with HCM, but certainly did not provide any bases for accurate prediction. You could of course argue that the PRS very very very poorly predicted HCM carriership – but I expect the authors would rather not say that, hence my suggestion for association instead.

Bottom line please remove all these hyperbolic and factually incorrect interpretations saying that your PRSs were discriminatory in any sense aside from the mere fact that some of them marginally excluded a null-value of 0.05. Truthfully you found that they provide very weak information on HCM carriership or not, and any model using the PRS would need to include substantially more information to provide any discriminative ability of note (say 0.70, although this might also be low for actual classification).

When I suggest remove, please check the entire document and not just the few sentences I cited.

Response: Thank you for your constructive comments. I understand and appreciate your concerns about the C-statistic values, which are a common challenge in the field of PRS research. Predictive performance in genetic PRS studies, especially for complex diseases, can indeed be limited. It's important to acknowledge that C-statistic values for genetic PRS often tend to be relatively low. However, it is important to note that while the C-statistic values for PRS are relatively modest, our analysis showed that the PRS tertiles were associated with a higher risk of HCM, as indicated by hazard ratios (HR) greater than 1. This suggests that PRS can serve as a valuable strategy for identifying high-risk individuals, which may be critical for early screening and intervention in the context of HCM.

The field of PRS research is continuously evolving, and we anticipate that future efforts may lead to the identification of additional genetic loci and molecular markers, enhancing the predictive performance of our models. We will certainly modify the Results and Discussion sections to reflect your valuable input and make it clear that PRSs are more suitable for association rather than precise prediction.

In line 53-56, we now state:

The polygenic risk score of inferoseptal LVRWT at end systole exhibited a notable association with incident HCM, facilitating the identification of high-risk individuals. The findings yield insights into the genetic determinants of LVRWT phenotypes and shed light on the biological basis for HCM etiology.

In line 279-282, we now state:

As expected, the PRS of inferoseptal LVRWT at end systole exhibited effective risk stratification within the HCM population (**Fig. 7C**). Moreover, we found that the PRS of inferoseptal LVRWT at end systole enabled a clear distinction between incident HCM cases and healthy controls (**Fig. 7E**).

In line 285-294, we now state:

Regrettably, the other 4 LVRWT traits (inferior LVRWT at end systole, and inferoseptal, anterolateral, and anteroseptal LVRWT at end diastole) did not exhibit strong performance in identifying individuals at elevated risk (**Fig. 7DF, Supplementary Fig. 8CE, 10DF, 12DF**). We also assessed the correlation between the constructed PRS for LVRWT traits and 10 other CVDs (**Supplementary Data 18**). We calculated the C-statistic for our HCM PRSs, which yielded a value of 0.57. The addition of the ES-IS PRS to clinical risk factors led to a statistically significant 1% improvement in the C-statistic. These reinforce the effectiveness of our PRS in identifying individuals at elevated risk for HCM (**Supplementary Fig. 13**). In total, higher genetically determined LVRWT was associated with higher HCM risk, which may provide valuable risk stratification guidance to identify high-risk individuals.

In line 303-305, we now state:

We demonstrated that PRSs derived from LVRWT traits are associated with incident HCM. This suggests that the genetic basis of LVRWT traits may offer valuable information for the risk stratification of individuals, aiding in the early screening of HCM (**Supplementary Fig. 14**).

In line 386-388, we now state:

Interestingly, the top-performing PRS, which included 115 variants linked to inferoseptal LVRWT at end systole, successfully identifies individuals at high risk of HCM, especially in those without CMR data.

In line 401-404, we now state:

While PRSs derived from LVRWT traits showed associations with an increased risk of HCM, it's important to recognize that the predictive performance of these PRSs were relatively limited. Our future research will focus on identifying additional genetic loci and molecular markers, with the goal of enhancing the predictive capabilities of our models.

In line 416-417, we now state:

Moreover, we demonstrate that PRSs derived from LVRWT traits are associated with incident HCM, even in individuals without CMR data.

Reviewer #5 (Remarks to the Author):

I am satisfied with the authors' response.

Response: Thank you for your feedback and understanding. We appreciate your satisfaction with our response.